# Synthetic Combinations:
# A Causal Framework for Combinatorial Interventions

**Abhineet Agarwal**
Department of Statistics
University of California, Berkeley
aa3797@berkeley.edu

**Anish Agarwal**[*]
Department of IEOR
Columbia University
aa5194@columbia.edu

**Suhas Vijaykumar**[†]
Amazon, Core AI
suhasv@mit.edu

## Abstract

We consider a setting where there are $N$ heterogeneous units and $p$ interventions. Our goal is to learn unit-specific potential outcomes for any combination of these $p$ interventions, i.e., $N \times 2^p$ causal parameters. Choosing a combination of interventions is a problem that naturally arises in a variety of applications such as factorial design experiments and recommendation engines (e.g., showing a set of movies that maximizes engagement). Running $N \times 2^p$ experiments to estimate the various parameters is likely expensive and/or infeasible as $N$ and $p$ grow. Further, with observational data there is likely confounding, i.e., whether or not a unit is seen under a combination is correlated with its potential outcome. We study this problem under a novel model that imposes latent structure across *both* units and combinations of interventions. Specifically, we assume latent similarity in potential outcomes across units (i.e., the matrix of potential outcomes is rank $r$) and regularity in how combinations of interventions interact (i.e., the coefficients in the Fourier expansion of the potential outcomes is $s$ sparse). We establish identification for all $N \times 2^p$ parameters despite unobserved confounding. We propose an estimation procedure, Synthetic Combinations, and establish finite-sample consistency under precise conditions on the observation pattern. We show that Synthetic Combinations is able to consistently estimate unit-specific potential outcomes given a total of $\text{poly}(r) \times (N + s^2 p)$ observations. In comparison, previous methods that do not exploit structure across both units and combinations have poorer sample complexity that scales as $\min(N \times s^2 p, \ \text{poly}(r) \times (N + 2^p))$.

## 1 Introduction

Modern-day decision makers—in settings from e-commerce to public policy to medicine—often must select a combination of actions, and would like to do so in a highly personalized manner. Examples include recommending a curated basket of items to customers on a commerce platform, deciding on a combination of therapies for a medical patient, enacting a collection of socio-economic policies for a specific geographic location, conjoint analysis in surveys, selecting important feature sets for machine learning models, etc. Despite the ubiquity of this setting, it comes with significant empirical challenges: with $p$ interventions and $N$ units, a decision maker must evaluate $N \times 2^p$ potential combinations in order to confirm the optimal personalized policy. With large $N$ and even with relatively small $p$ (due to exponential dependence), it becomes infeasible to run that many experiments; in observational data there is the additional challenge of potential unobserved confounding. Current methods tackle this problem by following one of two approaches: (i) they impose structure on how combinations of interventions interact, or (ii) they assume latent similarity

---

[*]Work done while post-doc at Amazon, Core AI .

[†]Work done while at MIT.

37th Conference on Neural Information Processing Systems (NeurIPS 2023).

in potential outcomes across units. However, as we discuss in detail below, these approaches require a large number of observations to estimate all $N \times 2^p$ potential outcomes because they do not exploit structure across both units and combinations. This motivates the question: *how can one effectively share information across both units and combinations of interventions?*

**Contributions.** Our contributions may be summarized as follows. **(1)** For a unit $n \in [N]$, we represent its potential outcomes over the $2^p$ combinations as a Boolean function from $\{-1,1\}^p$ to $\mathbb{R}$, expressed in the Fourier basis. To impose structure across combinations, we assume that for a unit $n$, the linear coefficients $\boldsymbol{\alpha}_n \in \mathbb{R}^{2^p}$ induced by this Fourier basis representation is $s$-sparse, i.e., has at most $s$ non-zero entries. To impose structure across units, we assume that this matrix of Fourier coefficients across units $\mathcal{A} = [\boldsymbol{\alpha}_n]_{n \in [N]} \in \mathbb{R}^{N \times 2^p}$ has rank $r$. This simultaneous sparsity and low-rank assumption allows the researcher to share information across both units and combinations. **(2)** We establish identification for the $N \times 2^p$ potential outcomes of interest, which requires that any confounding is mediated by the (unobserved) matrix of Fourier coefficients $\mathcal{A}$. **(3)** We design a two-step algorithm "Synthetic Combinations" and prove it consistently estimates the various causal parameters, despite potential unobserved confounding. The first step of Synthetic Combinations, termed "horizontal regression", learns the structure across combinations of interventions—assuming sparsity in the Fourier coefficients—via the Lasso. The second step, termed "vertical regression", learns the structure across units—assuming the matrix of Fourier coefficients are low-rank—via principal component regression (PCR). **(4)** We show that Synthetic Combinations is able to consistently estimate unit-specific potential outcomes given a total of $\text{poly}(r) \times (N + s^2 p)$ observations (ignoring logarithmic factors). This improves over previous methods that do not exploit structure across both units and combinations, which have sample complexity scaling as $\min(N \times s^2 p, \ \text{poly}(r) \times (N + 2^p))$. A summary of the sample complexities required for different methods can be found in Table 1. A key technical challenge in our proofs is analyzing how the error induced in the first step of Synthetic Combinations percolates through to the second step. To tackle it, we reduce the problem to that of high-dimensional error-in-variables regression with linear model misspecification, and do a novel analysis of this statistical setting.

## 2   Related Work

**Learning over Combinations.** To place structure on the space of combinations, we use tools from the theory of learning (sparse) Boolean functions, in particular the Fourier transform. Sparsity of the Fourier transform was proposed as a complexity measure to characterize and design learning algorithms for low-depth trees, low-degree polynomials, and small circuits [12, 26]. Learning Boolean functions is now a central topic in learning theory, and is closely related to many important questions in ML more broadly; see e.g. [31] for discussion of the $k$-Junta problem and its relation to relevant feature selection. We refer to O'Donnell [34, Chapter 3] for further background on this area. In this paper, we focus on [32] which showed that Lasso can be used to efficiently learn sparse Boolean functions, an essential property in our setting since the dimension of the function class grows exponentially.

**Matrix Completion.** We build on the observation that imputing counterfactual outcomes in the presence of a latent factor structure can be equivalently expressed as low-rank matrix completion [9, 8, 5]. The observation that low-rank matrices may typically be recovered from a small fraction of the entries by nuclear-norm minimization has had a major impact on modern statistics [15, 35, 13]. In the noisy setting, proposed estimators have generally proceeded by minimizing risk subject to a nuclear-norm penalty, such as in the SoftImpute algorithm of [30], or minimizing risk subject to a rank constraint as in the hard singular-value thresholding (HSVT) algorithms analyzed by [27, 28, 23, 16]. We refer the reader to [19, 33] for a comprehensive overview of this vast literature.

| Learning algorithm | Exploits structure across combinations ($\|\boldsymbol{\alpha}_n\|_0 = s$) | Exploits structure across units (rank($\mathcal{A}$) = $r$) | Sample complexity |
|---|---|---|---|
| Lasso | ✓ | ✗ | $O(N \times s^2 p)$ |
| Matrix Completion | ✗ | ✓ | $O(\text{poly}(r) \times (N + 2^p))$ |
| Synthetic Combinations | ✓ | ✓ | $O(\text{poly}(r) \times (N + s^2 p))$ |

Table 1: Comparison of sample complexity of Synthetic Combinations to other methods.

**Econometrics/causal inference.** There is a rich literature on how to learn personalized treatment effects for heterogeneous units. This problem is of particular importance in the social sciences and in medicine, where experimental data is limited, and has led to several promising approaches including instrumental variables, difference-in-differences, regression discontinuity, and others; see [7, 25] for an overview. Of particular interest to us here is the "synthetic control" method [1], which exploits an underlying factor structure to effectively "share" counterfactual information between treatment and control units. Building on [1], recent work has shown that the same underlying structure can be used to estimate treatment effects with multiple treatments *and* heterogeneous units despite unmeasured confounding. The resulting framework, called "synthetic interventions" (SI), uses the factor representation to efficiently share information across similar (yet distinct) units and treatments [5]. We generalize the SI framework to settings where treatments are combinations of interventions and most treatments have no units that receive it, but there is structure across combinations of interventions. In doing so, we enable its use in highly practical cases where multiple interventions are delivered simultaneously, such as recommender systems, medical treatment regimens, and factorial design experiments.

## 3 Setup and Model

In this section, we first describe requisite notation, background on the Fourier expansion of real-valued functions over booleans, and how it relates to potential outcomes over combinations.

### 3.1 Notation

**Representation of Combinations as Binary Vectors**. Let $[p] = \{1, \dots p\}$ denote the set of $p$ interventions. Denote by $\Pi$ the power set of $[p]$, i.e., the set of all possible combinations of $p$ interventions, where we note $|\Pi| = 2^p$. Then, any given combination $\pi \in \Pi$ induces the following binary representation $\mathbf{v}(\pi) \in \{-1, 1\}^p$ defined as follows: $\mathbf{v}(\pi)_i = 2\,\mathbf{1}\{i \in \pi\} - 1$.

**Fourier Expansion of Boolean Functions.** Let $\mathcal{F}_{\text{bool}} = \{f : \{-1, 1\}^p \to \mathbb{R}\}$ be the set of all real-valued functions defined on the hypercube $\{-1, 1\}^p$. Then $\mathcal{F}_{\text{bool}}$ forms a Hilbert space defined by the following inner product: for any $f, g \in \mathcal{F}_{\text{bool}}$, $\langle f, g \rangle_B = \frac{1}{2^p} \sum_{\mathbf{x} \in \{-1, 1\}^p} f(\mathbf{x}) g(\mathbf{x})$. This inner product induces the norm $\langle f, f \rangle_B := \|f\|_B^2 = \frac{1}{2^p} \sum_{\mathbf{x} \in \{-1, 1\}^p} f^2(\mathbf{x})$. We construct an orthonormal basis for $\mathcal{F}_{\text{bool}}$ as follows: for each subset $S \subset [p]$, define a basis function $\chi_S(\mathbf{x}) = \prod_{i \in S} x_i$ where $x_i$ is the $i^{\text{th}}$ coefficient of $\mathbf{x} \in \{-1, 1\}^p$. One can verify that for any $S \subset [p]$ that $\|\chi_S\|_B = 1$, and that $\langle \chi_S, \chi_{S'} \rangle_B = 0$ for any $S' \neq S$. Since $|\{\chi_S : S \subset [p]\}| = 2^p$, the functions $\chi_S$ are an orthonormal basis of $\mathcal{F}_{\text{bool}}$.

Hence, any $f \in \mathcal{F}_{\text{bool}}$ can be expressed via the following "Fourier" decomposition: $f(\mathbf{x}) = \sum_{S \subset [p]} \alpha_S \chi_S(\mathbf{x})$, where the Fourier coefficient $\alpha_S$ is given by computing $\alpha_S = \langle f, \chi_S \rangle_B$. We will refer to $\chi_S$ as the Fourier character. Define $\boldsymbol{\alpha}_f = [\alpha_S]_{S \in [p]} \in \mathbb{R}^{2^p}$ and $\boldsymbol{\chi}(x) = [\chi_S(\mathbf{x})]_{S \in [p]} \in \{-1, 1\}^{2^p}$ as the vector of Fourier coefficients and characters respectively. Hence any function $f : \{-1, 1\}^p \to \mathbb{R}$ can be re-expressed as follows: $f(\mathbf{x}) = \langle \boldsymbol{\alpha}_f, \boldsymbol{\chi}(x) \rangle$. For $\pi \in \Pi$, abbreviate $\chi_S(\mathbf{v}(\pi))$ and $\boldsymbol{\chi}(\mathbf{v}(\pi))$ as $\chi_S^\pi$ and $\boldsymbol{\chi}^\pi$ respectively.

**Observed and potential outcomes.** Let $Y_n^{(\pi)} \in \mathbb{R}$ denote the *potential outcome* for unit $n$ under combination $\pi$ and $Y_{n\pi} \in \{\mathbb{R} \cup \star\}$ as the *observed outcome*, where $\star$ indicates a missing value, i.e., the outcome associated with the unit-combination pair $(n, \pi)$ was not observed. Let $\mathbf{Y} = [Y_{n\pi}] \in \{\mathbb{R} \cup \star\}^{N \times 2^p}$. Let $\mathcal{D} \subset [N] \times [2^p]$, refer to the subset of unit-combination pairs we do observe, i.e.,

$$Y_{n\pi} = \begin{cases} Y_n^{(\pi)}, & \text{if } (n, \pi) \in \mathcal{D} \\ \star, & \text{otherwise.} \end{cases} \tag{1}$$

Note that (1) implies stable unit treatment value assignment (SUTVA) holds. Let $\Pi_S \subseteq \Pi$ denote a subset of combinations. For a given unit $n$, let $\mathbf{Y}_{\Pi_S, n} = [Y_{n\pi_i} : \pi_i \in \Pi_S] \in \{\mathbb{R} \cup \star\}^{|\Pi_S|}$ represent the vector of observed outcomes for all $\pi \in \Pi_S$. Similarly, let $\mathbf{Y}_n^{(\Pi_S)} = [Y_n^{(\pi_i)} : \pi_i \in \Pi_S] \in \mathbb{R}^{|\Pi_S|}$ represent the vector of potential outcomes. Denote $\boldsymbol{\chi}(\Pi_S) = [\boldsymbol{\chi}^{\pi_i} : \pi_i \in \Pi_S] \in \{-1, 1\}^{|\Pi^S| \times 2^p}$.

## 3.2 Model & Target Causal Parameter

Define $Y_n^{(\cdot)}: \pi \to \mathbb{R}$ as a real-valued function over the hypercube $\{-1,1\}^p$ associated with unit $n$. It takes as input a combination $\pi$, converts it to a $p$-dimensional binary vector $\mathbf{v}(\pi)$, and outputs a real number $Y_n^{(\pi)}$. Given the discussion in Section 3.1, it follows that $Y_n^{(\pi)}$ always has the representation $\langle \boldsymbol{\alpha}_n, \boldsymbol{\chi}^\pi \rangle$ for some $\boldsymbol{\alpha}_n \in \mathbb{R}^{2^p}$. Thus, without any loss of generality, the $\boldsymbol{\alpha}_n$ are unit-specific latent variables (Fourier coefficients) encoding the treatment response function. Below, we state our key assumption on these induced Fourier coefficients.

**Assumption 3.1** (Potential Outcome Model). *For any unit-combination pair $(n,\pi)$, we assume it has the following representation,*

$$Y_n^{(\pi)} = \langle \boldsymbol{\alpha}_n, \boldsymbol{\chi}^\pi \rangle + \epsilon_n^\pi, \tag{2}$$

*where $\boldsymbol{\alpha}_n \in \mathbb{R}^{2^p}$ and $\boldsymbol{\chi}^\pi \in \{-1,1\}^{2^p}$ are the Fourier coefficients and characters, respectively. We assume the following properties: (a) low-rank: the matrix $\mathcal{A} = [\boldsymbol{\alpha}_n]_{n \in [N]}$ has rank $r \in [\min\{N, 2^p\}]$; (b) sparsity: $\boldsymbol{\alpha}_n$ is $s$-sparse (i.e. $\|\boldsymbol{\alpha}_n\|_0 \leq s$, where $s \in [2^p]$) for every unit $n \in [N]$; (c) $\epsilon_n^\pi$ is a residual term specific to $(n,\pi)$ which satisfies $\mathbb{E}[\epsilon_n^\pi \mid \mathcal{A}] = 0$.*

The assumption is then that each $\boldsymbol{\alpha}_n$ $s$-sparse and $\mathcal{A}$ is rank-$r$; $\epsilon_n^\pi$ is the residual from this sparse and low-rank approximation, and it serves as the source of uncertainty in our model. Given $\mathbb{E}[\epsilon_n^\pi \mid \mathcal{A}] = 0$, the matrix $\mathbb{E}[\mathbf{Y}_N^{(\Pi)}] = [\mathbb{E}[\mathbf{Y}_n^{(\Pi)}] : n \in [N]] \in \mathbb{R}^{2^p \times N}$ also is rank $r$, where the expectation is defined with respect to $\epsilon_n^\pi$. This is because $\mathbb{E}[\mathbf{Y}_N^{(\Pi)}]$ can be written as $\mathbb{E}[\mathbf{Y}_N^{(\Pi)}] = \boldsymbol{\chi}(\Pi)\mathcal{A}^T$, and since $\boldsymbol{\chi}(\Pi)$ is an invertible matrix, $\text{rank}(\mathbb{E}[\mathbf{Y}_N^{(\Pi)}]) = \text{rank}(\mathcal{A})$. The low-rank property places *structure across units*; that is, we assume there is sufficient similarity across units so that $\mathbb{E}[\mathbf{Y}_n^{(\Pi)}]$ for any unit $n$ can be written as a linear combination of $r$ other rows of $\mathbb{E}[\mathbf{Y}_N^{(\Pi)}]$. This is a standard assumption used to encode latent similarity across units in matrix completion and its related applications (e.g., recommendation engines).

Sparsity establishes *unit-specific* structure; that is, we assume that the potential outcomes for a given user only depend on a small subset of the functions $\{\chi_S : S \subset [p]\}$. We emphasize that this subset of functions can be different across units. As discussed in Section 2, sparsity is commonly employed when studying the learnability of Boolean functions. In the context of recommendation engines, sparsity is implied if the ratings for a set of goods only depend on a small number of combinations of items within that set. Sparsity is also often assumed implicitly in factorial design experiments, where analysts typically only include pairwise interaction effects between interventions and ignore higher-order interactions [24]. We discuss further applications of combinatorial inference in greater detail in Appendix A.

Next, we present an assumption that formalizes the dependence (i.e., confounding) between the missingness pattern induced by the treatment assignments $\mathcal{D}$ and the potential outcomes $Y_n^{(\pi)}$, and provide an interpretation of the induced data generating process (DGP) for potential outcomes.

**Assumption 3.2** (Selection on Fourier coefficients). *For all $n \in [N]$ and $\pi \in \Pi$, $Y_n^{(\pi)} \perp\!\!\!\perp \mathcal{D} \mid \mathcal{A}$.*

**DGP.** Given Assumptions 3.1 and Assumption 3.2, the DGP can be summarized as follows: (i) unit-specific latent Fourier coefficients $\mathcal{A}$ are either deterministic or sampled from an unknown distribution; we will condition on this quantity throughout. (ii) Given $\mathcal{A}$, we sample mean-zero random variables $\epsilon_n^\pi$, and generate potential outcomes according to our model $Y_n^{(\pi)} = \langle \boldsymbol{\alpha}_n, \boldsymbol{\chi}^\pi \rangle + \epsilon_n^\pi$. (iii) $\mathcal{D}$ is allowed to depend on unit-specific latent Fourier coefficients $\mathcal{A}$ (i.e. $\mathcal{D} = f(\mathcal{A})$). We define all expectations w.r.t. noise, $\epsilon_n^\pi$. This DGP introduces unobserved confounding since $Y_n^{(\pi)} \not\!\perp\!\!\!\perp \mathcal{D}$. However, this DGP does imply that Assumption 3.2 holds, i.e., conditional on the Fourier coefficients $\mathcal{A}$, the potential outcomes are independent of the treatment assignments $\mathcal{D}$. This conditional independence condition can be thought of as "selection on latent Fourier coefficients", which generalizes the widely made assumption of "selection on observables." The latter requires that potential outcomes are independent of treatment assignments conditional on *observed* covariates–we reemphasize that $\mathcal{A}$ is unobserved.

**Target parameter.** For any unit-combination pair $(n,\pi)$, we aim to estimate $\mathbb{E}[Y_n^{(\pi)} \mid \mathcal{A}]$, where the expectation is w.r.t. $\epsilon_n^\pi$, and we condition on the set of Fourier coefficients $\mathcal{A}$.

# 4 Identification of Potential Outcomes

We show that $\mathbb{E}[Y_n^{(\pi)} \mid \mathcal{A}]$ can be written as a function of observed outcomes, i.e., we establish identification of our target causal parameter. As discussed earlier, our model allows for *unobserved confounding*: whether or not a unit is seen under a combination may be correlated with its potential outcome under that combination due to unobserved factors, as long as certain conditions are met. We introduce necessary notation and assumption required for our result. For a unit $n \in [N]$, denote the subset of combinations we observe them under as $\Pi_n \subseteq \Pi$. For $\pi \in \Pi$, let $\tilde{\chi}_n^\pi \in \mathbb{R}^{2^p}$ denote the vector where we zero out all coordinates of $\chi^\pi \in \mathbb{R}^{2^p}$ that correspond to the coefficients of $\boldsymbol{\alpha}_n$ which are zero. For example, if $\boldsymbol{\alpha}_n = (1,1,0,0,...0)$ and $\chi^\pi = (1,1,...1)$, then $\tilde{\chi}_n^\pi = (1,1,0,...0)$. We the make the following assumption.

**Assumption 4.1** (Donor Units). *We assume there exists a set of "donor units" $\mathcal{I} \subset [N]$, such that the following two conditions hold:*

(a) *Horizontal span inclusion: For any donor unit $u \in \mathcal{I}$ and combination $\pi \in \Pi$, suppose $\tilde{\chi}_u^\pi \in span(\tilde{\chi}_u^{\pi_i} : \pi_i \in \Pi_u)$. That is, there exists exists $\boldsymbol{\beta}_{\Pi_u}^\pi \in \mathbb{R}^{|\Pi_u|}$ such that $\tilde{\chi}_u^\pi = \sum_{\pi_i \in \Pi_u} \beta_{\pi_i}^\pi \tilde{\chi}_u^{\pi_i}$.*

(b) *Linear span inclusion: For any unit $n \in [N] \setminus \mathcal{I}$, suppose $\boldsymbol{\alpha}_n \in span(\boldsymbol{\alpha}_u : u \in \mathcal{I})$. That is, there exists $\mathbf{w}^n$ such that $\boldsymbol{\alpha}_n = \sum_{u \in \mathcal{I}} w_u^n \boldsymbol{\alpha}_u$*

Horizontal span inclusion requires that the set of observed combinations for any donor unit is "diverse" enough that the projection of the Fourier characteristic for a target intervention is in the span of characteristics of observed interventions. Linear span inclusion requires that the donor set is diverse enough such that the Fourier coefficient of any unit is in the span of the Fourier coefficients of the donor set.

## 4.1 Identification Result

Given these assumptions, we now present our identification theorem.

**Theorem 4.2.** *Let Assumptions 3.1, 3.2, 4.1 hold. Given $\boldsymbol{\beta}_{\Pi_u}^\pi$ and $\mathbf{w}_u^n$ defined in Assumption 4.1, we have*

(a) *Donor units: For $u \in \mathcal{I}$, and $\pi \in \Pi$, $\mathbb{E}[Y_u^{(\pi)} \mid \mathcal{A}] = \sum_{\pi_u \in \Pi_u} \beta_{\pi_u}^\pi \mathbb{E}[Y_{u,\pi_u} \mid \mathcal{A}, \mathcal{D}]$.*

(b) *Non-donor units: For $n \in [N] \setminus \mathcal{I}$, and $\pi \notin \Pi_n$, $\mathbb{E}[Y_n^{(\pi)} \mid \mathcal{A}] = \sum_{u \in \mathcal{I}, \pi_u \in \Pi_u} w_u^n \beta_{\pi_u}^\pi \mathbb{E}[Y_{u,\pi_u} \mid \mathcal{A}, \mathcal{D}]$.*

Theorem 4.2 gives conditions under which the donor set $\mathcal{I}$ and the observation pattern $\mathcal{D}$ are sufficient to recover the full set of unit specific potential outcomes $\mathbb{E}[Y_n^{(\pi)} | \mathcal{A}]$ in the noise-free limit. Part (a) establishes that for every donor unit $u \in \mathcal{I}$, the causal estimand can be written as a function of its *own* observed outcomes $\mathbb{E}[\mathbf{Y}_{\Pi_u}]$, given knowledge of $\boldsymbol{\beta}_{\Pi_u}^\pi$. Part (b) states that the target causal estimand $\mathbb{E}[Y_n^{(\pi)}]$ for a non-donor unit and combination $\pi$ can be written as a linear combination of the outcomes of the donor set $\mathcal{I}$, given knowledge of $\mathbf{w}_n^n$. Previous works that establish identification under a latent factor model requires a growing number of donor units to be observed under all treatments [6]. This is infeasible in our setting because *the vast majority of combinations have no units that receive it*. As a result, we have to first to identify the outcomes of donor units under all combinations (part (a)), before transferring them to non-donor units (part (b)). In order to do so, Theorem 4.2 suggests that the key quantities in estimating $\mathbb{E}[Y_n^{(\pi)}]$ for any unit-combination pair $(n,\pi)$ are $\boldsymbol{\beta}_{\Pi_u}^\pi$ and $\mathbf{w}_u^n$. In the following section, we provide an algorithm to estimate both $\boldsymbol{\beta}_{\Pi_u}^\pi$ and $\mathbf{w}_u^n$, as well as concrete ways of determining the donor set $\mathcal{I}$.

# 5 The Synthetic Combinations Estimator

We now describe the Synthetic Combinations estimator, a simple and flexible two-step procedure for estimating our target causal parameter. See Figure 1 for a pictorial representation.

**Step 1: Horizontal Regression.** We denote the vector of observed responses $\mathbf{Y}_{n,\Pi_n} = [Y_{n\pi} : \pi \in \Pi_n] \in \mathbb{R}^{|\Pi_n|}$ for any unit $n$ as $\mathbf{Y}_{\Pi_n}$. Then, for every unit $u$ in the donor set $\mathcal{I}$, we estimate $\mathbb{E}[Y_u^{(\pi)}]$

via the Lasso, i.e., by solving the following convex program with penalty parameter $\lambda_u$:

$$\hat{\boldsymbol{\alpha}}_u = \underset{\boldsymbol{\alpha}}{\operatorname{argmin}} \; \frac{1}{|\Pi_u|} \|\mathbf{Y}_{\Pi_u} - \boldsymbol{\chi}(\Pi_u)\boldsymbol{\alpha}\|_2^2 + \lambda_u \|\boldsymbol{\alpha}\|_1 \tag{3}$$

where recall that $\boldsymbol{\chi}(\Pi_u) = [\boldsymbol{\chi}^\pi : \pi \in \Pi_u] \in \mathbb{R}^{|\Pi_u| \times 2^p}$. Then, for any donor unit-combination pair $(u,\pi)$, let $\hat{\mathbb{E}}[Y_u^{(\pi)}] = \langle \hat{\boldsymbol{\alpha}}_u, \boldsymbol{\chi}^\pi \rangle$ denote the estimate of the potential outcome $\mathbb{E}[Y_u^{(\pi)}]$.

**Step 2: Vertical Regression.** Next, we estimate potential outcomes for all units $n \in [N] \setminus \mathcal{I}$. To do so, we define some required notation. For $\Pi_S \subseteq \Pi$, define the vector of estimated potential outcomes $\hat{\mathbb{E}}[\mathbf{Y}_u^{(\Pi_S)}] = [\hat{\mathbb{E}}[Y_u^{(\pi)}] : \pi \in \Pi^S] \in \mathbb{R}^{|\Pi_S|}$. Additionally, let $\hat{\mathbb{E}}[\mathbf{Y}_{\mathcal{I}}^{(\Pi_S)}] = [\hat{\mathbb{E}}[\mathbf{Y}_u^{(\Pi_S)}] : u \in \mathcal{I}] \in R^{|\Pi_S| \times |\mathcal{I}|}$.

*Step 2(a): Principal Component Regression.* Perform a singular value decomposition (SVD) of $\hat{\mathbb{E}}[\mathbf{Y}_{\mathcal{I}}^{(\Pi_n)}]$ to get $\hat{\mathbb{E}}[\mathbf{Y}_{\mathcal{I}}^{(\Pi_n)}] = \sum_{l=1}^{\min(|\Pi_n|,|\mathcal{I}|)} \hat{s}_l \hat{\boldsymbol{\mu}}_l \hat{\boldsymbol{\nu}}_l^T$. Using a hyper-parameter $\kappa \leq \min(|\Pi_n|,|\mathcal{I}|)^3$, compute $\hat{\mathbf{w}}^n \in \mathbb{R}^{|\mathcal{I}|}$ as follows:

$$\hat{\mathbf{w}}^n = \left( \sum_{l=1}^{\kappa} \hat{s}_l^{-1} \hat{\boldsymbol{\nu}}_l \hat{\boldsymbol{\mu}}_l^T \right) \mathbf{Y}_{\Pi_n} \tag{4}$$

*Step 2(b): Estimation.* Using $\hat{\mathbf{w}}^n = [\hat{w}_u^n : u \in \mathcal{I}]$, we have the following estimate for any $\pi \in \Pi$

$$\hat{\mathbb{E}}[Y_n^{(\pi)}] = \sum_{u \in \mathcal{I}} \hat{w}_u^n \hat{\mathbb{E}}[Y_u^{(\pi)}] \tag{5}$$

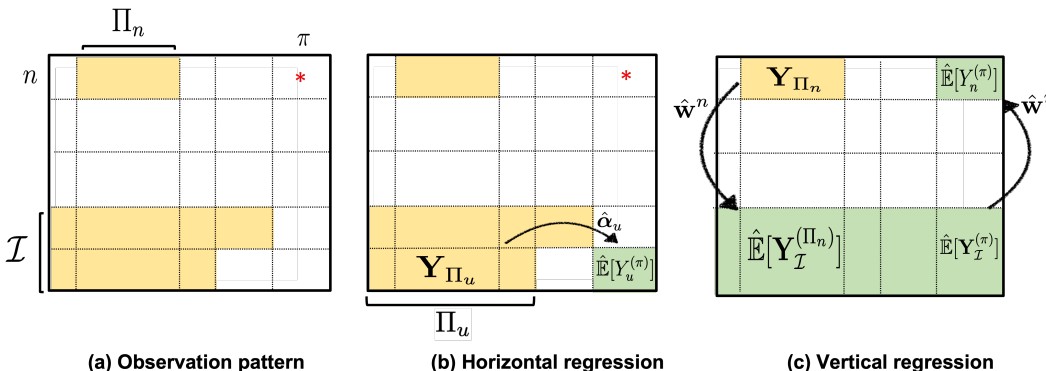

**(a) Observation pattern**      **(b) Horizontal regression**      **(c) Vertical regression**

Figure 1: (a) depicts an example of a particular observation pattern with outcome for unit-combination pair $(n,\pi)$ missing. (b) demonstrates horizontal regression for donor unit $u$ where we estimate potential outcome $\mathbb{E}[Y_u^{(\pi)}]$. (c) visualizes vertical regression where we transfer estimated outcomes from the donor set $\mathcal{I}$ to $(n,\pi)$.

**Suitability of Lasso and PCR.** Lasso is appropriate for the horizontal regression because $\boldsymbol{\alpha}_u$ is sparse. However, Synthetic Combinations allows for any ML algorithm (e.g., neural networks, random forests) to be used in the first step. This flexibility allows an analyst to tailor the horizontal learning procedure, and include prior information. However, we leave this model-agnostic analysis as future work, and focus on the Lasso for simplicity. PCR is appropriate for the vertical regression because $\mathcal{A}$ is low rank. As [3, 4] show, PCR implicitly regularizes the regression by adapting to the rank of the covariates $(\mathbf{Y}_{\Pi_n})$, i.e., the out-of-sample error of PCR scales with $r$ rather than the ambient covariate dimension.

**Determining Donor Set $\mathcal{I}$.** Synthetic Combinations requires the existence of a subset of units $\mathcal{I} \subset [N]$ such that we are able to (i) accurately estimate their potential outcomes under all possible combinations, and (ii) transfer these estimated outcomes to a unit $n \in [N] \setminus \mathcal{I}$. Theoretically, we detail sufficient conditions on the observation pattern such that we are able to perform (i) and (ii) accurately via the Lasso and PCR respectively. In practice, we recommend the following to determine $\mathcal{I}$. For every unit $n \in [N]$, learn a separate Lasso model $\boldsymbol{\alpha}_n$ and assess its performance through cross-validation (CV). Assign units with low CV error (with a pre-determined threshold) as the donor set $\mathcal{I}$,

---

[3]Both $\lambda$ and $\kappa$ can be chosen in a data-driven manner (e.g., via CV) as discussed in [17] and [4] respectively.

and estimate outcomes $\hat{\mathbb{E}}[Y_u^{(\pi)}]$ for every unit $u \in \mathcal{I}$ and $\pi \in \Pi$. For non-donor units, PCR performance can also be assessed via k-fold CV. For units with low PCR error, linear span inclusion (Assumption 4.1(b)) and the assumptions required for the generalization for PCR likely hold, and hence we estimate their potential outcomes as in (5). For units with large PCR error, it is either unlikely that these set of assumptions holds or that $|\Pi_n|$ is not large enough (i.e., additional experiments need to be run for this unit), and hence we do not recommend estimating their counterfactuals.

# 6 Synthetic Combinations Theoretical Analysis

In this section, we establish finite-sample consistency of Synthetic Combinations, starting with a discussion of the additional assumptions required for our results.

## 6.1 Additional Assumptions

**Assumption 6.1** (Bounded Potential Outcomes). *We assume that* $\mathbb{E}[Y_n^{(\pi)}] \in [-1, 1]$ *for any unit-combination pair* $(n, \pi)$.

**Assumption 6.2** (Sub-Gaussian Noise). *Conditioned on* $\mathcal{A}$, *for any unit-combination pair* $(n, \pi)$, $\epsilon_n^\pi$ *are independent zero-mean sub-Gaussian random variables with* $\mathrm{Var}[\epsilon_n^\pi \mid \mathcal{A}] \leq \sigma^2$ *and* $\|\epsilon_n^\pi \mid \mathcal{A}\|_{\psi_2} \leq C\sigma$ *for some constant* $C > 0$.

**Assumption 6.3** (Incoherence of Donor Fourier characteristics). *For every unit* $u \in \mathcal{I}$, *assume* $\boldsymbol{\chi}(\Pi_u)$ *satisfies incoherence:* $\left\| \frac{\boldsymbol{\chi}(\Pi_u)^T \boldsymbol{\chi}(\Pi_u)}{|\Pi_u|} - \mathbf{I}_{2^p} \right\|_\infty \leq \frac{C'}{s}$, *for a universal constant* $C' > 0$.

To define our next set of assumptions, we introduce necessary notation. For any subset of combinations $\Pi_S \subset \Pi$, let $\mathbb{E}[\mathbf{Y}_{\mathcal{I}}^{(\Pi_S)}] = [\mathbb{E}[\mathbf{Y}_u^{(\Pi_S)}] : u \in \mathcal{I}] \in \mathbb{R}^{|\Pi_S| \times |\mathcal{I}|}$.

**Assumption 6.4** (Donor Unit Balanced Spectrum). *For a given unit* $n \in [N] \setminus \mathcal{I}$, *let* $r_n$ *and* $s_1 ... s_{r_n}$ *denote the rank and non-zero singular values of* $\mathbb{E}[\mathbf{Y}_{\mathcal{I}}^{(\Pi_n)} \mid \mathcal{A}]$. *We assume that the singular values are well-balanced, i.e., for universal constants* $c, c' > 0$, *we have that* $s_{r_n}/s_1 \geq c$, *and* $\|\mathbb{E}[\mathbf{Y}_{\mathcal{I}}^{(\Pi_n)} \mid \mathcal{A}]\|_F^2 \geq c'|\Pi_n||\mathcal{I}|$.

**Assumption 6.5** (Subspace Inclusion). *For a given unit* $n \in [N] \setminus \mathcal{I}$ *and intervention* $\pi \in \Pi \setminus \Pi_n$, *assume that* $\mathbb{E}[\mathbf{Y}_{\mathcal{I}}^{(\pi)}]$ *lies within the row-span of* $\mathbb{E}[\mathbf{Y}_{\mathcal{I}}^{(\Pi_n)}]$

Assumption 6.3 is necessary for finite-sample consistency when estimating $\boldsymbol{\alpha}_n$ via the Lasso estimator (3), and is commonly made when studying the Lasso [36]. Incoherence can also seen as a inclusion criteria for a unit $n$ to be included in the donor set $\mathcal{I}$. Assumption 6.3 for example holds (with high probability) if $\Pi_u$ is chosen uniformly at random and grows as $\omega(s^2 p)$ as shown in Lemma 2 of [32]. Assumption 6.4 requires that the non-zero singular values of $\mathbb{E}[\mathbf{Y}_{\mathcal{I}}^{(\Pi_n)} \mid \mathcal{A}]$ are well-balanced. This assumption is standard when studying PCR [3, 5], and within the econometrics literature [10, 22]. It can also be empirically validated by plotting the spectrum of $\hat{\mathbb{E}}[\mathbf{Y}_{\mathcal{I}}^{(\Pi_n)}]$; if the singular spectrum of $\hat{\mathbb{E}}[\mathbf{Y}_{\mathcal{I}}^{(\Pi_n)}]$ displays a natural elbow point, then Assumption 6.4 is likely. Assumption 6.5 is also commonly made when analyzing PCR [4, 5, 2]. It can be thought of as a "causal transportability" condition from the model learnt using $\Pi_n$ to the interventions $\pi \in \Pi \setminus \Pi_n$. That is, subspace inclusion allows us to generalize well, and accurately estimate $\mathbb{E}[Y_n^{(\pi)} \mid \mathcal{A}]$ using $\langle \hat{\mathbb{E}}[\mathbf{Y}_{\mathcal{I}}^{(\Pi_n)}], \hat{\mathbf{w}}^n \rangle$.

## 6.2 Finite Sample Consistency

The following result establishes finite-sample consistency of Synthetic Combinations. Without loss of generality, we will focus on estimating the pair of quantities $(\mathbb{E}[Y_u^{(\pi)}], \mathbb{E}[Y_n^{(\pi)}])$ for a given donor unit $u \in \mathcal{I}$, and non-donor unit $n \in [N] \setminus \mathcal{I}$ under treatment assignment $\pi \in \Pi$. To simplify notation, we will use $O_p$ notation: for any sequence of random vectors $X_n$, $X_n = O_p(\gamma_n)$ if, for any $\epsilon > 0$, there exists constants $c_\epsilon$ and $n_\epsilon$ such that $\mathbb{P}(\|X_n\|_2 \geq c_\epsilon \gamma_n) \leq \epsilon$ for every $n \geq n_\epsilon$. Similarly, we define $\tilde{O}_p(\gamma_n)$ which suppresses logarithmic terms. We will further absorb dependencies on $\sigma$ into $\tilde{O}_p(\cdot)$.

**Theorem 6.6** (Finite Sample Consistency of Synthetic Combinations). *Conditioned on* $\mathcal{A}$, *let Assumptions 3.1–4.1, and 6.1–6.5 hold. Then, the following statements hold.*

(a) *For the given donor unit-combination pair $(u, \pi)$, let the Lasso regularization parameter satisfy $\lambda_u = \Omega(\sqrt{\frac{p}{|\Pi_u|}})$. Then, we have that:* $|\hat{\mathbb{E}}[Y_u^{(\pi)}] - \mathbb{E}[Y_u^{(\pi)}]| = \tilde{O}_p\left(\sqrt{\frac{s^2 p}{|\Pi_u|}}\right)$.

(b) *For the given unit-combination pair $(n, \pi)$ where $n \in [N] \setminus \mathcal{I}$, let $\kappa = rank(\mathbb{E}[\mathbf{Y}_{\mathcal{I}}^{(\Pi_n)}]) := r_n$. Then, provided that $\min_{u \in \mathcal{I}} |\Pi_u| := M = \omega(r_n^2 s^2 p)$, we have that:*

$$\left|\hat{\mathbb{E}}[Y_n^{(\pi)}] - \mathbb{E}[Y_n^{(\pi)}]\right| = \tilde{O}_p\left(\frac{r_n^2 \sqrt{s^2 p}}{\sqrt{M \times \min\{|\Pi_n|, |\mathcal{I}|\}}} + \frac{r_n^2 s^2 p \sqrt{|\mathcal{I}|}}{M} + \frac{r_n}{|\Pi_n|^{1/4}}\right)$$

Establishing Theorem 6.6 requires a novel analysis of error-in-variables (EIV) linear regression. Specifically, the general EIV linear model is as follows: $Y = \mathbf{X}\beta + \epsilon$, $\mathbf{Z} = \mathbf{X} + \mathbf{H}$, where $Y$ and $\mathbf{Z}$ are observed. In our case, $Y = \mathbf{Y}_{\Pi_n}$, $\mathbf{X} = \mathbb{E}[\mathbf{Y}_{\mathcal{I}}^{(\Pi_n)}]$, $\beta = \mathbf{w}^n$, $\mathbf{Z} = \hat{\mathbb{E}}[\mathbf{Y}_{\mathcal{I}}^{(\Pi_n)}]$, and $\mathbf{H}$ is the error arising in estimating $\mathbb{E}[\mathbf{Y}_{\mathcal{I}}^{(\Pi_n)}]$ via the Lasso. Typically one assumes that $\mathbf{H}$ is is a matrix of independent sub-gaussian noise. Our analysis requires a novel worst-case analysis of $\mathbf{H}$ (due to the 2-step regression of Lasso and then PCR), in which each entry of is $\mathbf{H}$ simply bounded.

Next, we describe the conditions placed on $\Pi_n$, $M$, $\mathcal{I}$ to achieve consistent estimation of $(\mathbb{E}[Y_u^{(\pi)}], \mathbb{E}[Y_n^{(\pi)}])$. To simplify the discussion, we assume that $\min\{|\Pi_n|, |\mathcal{I}|\} = |\mathcal{I}|$. This condition can be enforced in practice by simply picking a subset of donor units such that $|\mathcal{I}| \leq |\Pi_n|$ when performing PCR (i.e., step 2 of Synthetic Combinations). Given this assumption, it can be verified that $|\Pi_n|$ and $M$ need to scale as $\omega(r_n^4)$ and $\omega(r_n^4 s^2 p)$ respectively to achieve $\max\left(|\hat{\mathbb{E}}[Y_u^{(\pi)}] - \mathbb{E}[Y_u^{(\pi)}]|, |\hat{\mathbb{E}}[Y_n^{(\pi)}] - \mathbb{E}[Y_n^{(\pi)}]|\right) = \tilde{o}_p(1)$. Next, we present a corollary that discusses how quickly the parameters $r_n$, $s$, and $p$ can grow with the number of observations $|\Pi_n|$ and $M$.

**Corollary 6.7.** *With the set-up of Theorem 6.6, if (a) $r_n = o(|\Pi_n|^{1/4})$ and (b) $s = o\left(\sqrt{M/(p\max\{|\Pi_n|, |\mathcal{I}|\})}\right)$ then $\max\left(|\hat{\mathbb{E}}[Y_u^{(\pi)}] - \mathbb{E}[Y_u^{(\pi)}]|, |\hat{\mathbb{E}}[Y_n^{(\pi)}] - \mathbb{E}[Y_n^{(\pi)}]|\right) = \tilde{o}_p(1)$ as $M, |\Pi_n|, |\mathcal{I}| \to \infty$.*

Corollary 6.7 quantifies how $s, r_n$ can scale with the number of observations to achieve consistency. That is, Corollary 6.7 reflects the maximum "complexity" allowed for a given sample size.

### 6.3 Sample Complexity

We discuss the sample complexity of Synthetic Combinations to estimate all $N \times 2^p$ causal parameters, and compare it to that of other methods. To ease our discussion, we will ignore dependence on logarithmic factors and $\sigma$. Even if potential outcomes $Y_n^{(\pi)}$ were observed for all unit-combination pairs, consistently estimating $\mathbb{E}[\mathbf{Y}_N^{(\Pi)}]$ is not trivial. This is because, we only get to observe a single and noisy version $Y_{n\pi} = \langle \boldsymbol{\alpha}_n, \boldsymbol{\chi}^\pi \rangle + \epsilon_n^\pi$. Hypothetically, if we observe $K$ independent samples of $Y_{n\pi}$ for a given $(n, \pi)$, denoted by $Y_{n\pi}^1, ..., Y_{n\pi}^K$, the maximum likelihood estimator would be the empirical average $\frac{1}{K}\sum_{i=1}^K Y_{n\pi}^i$. The empirical average would concentrate around $\mathbb{E}[Y_n^{(\pi)}]$ at a rate $O(1/\sqrt{K})$ and hence would require $K = \Omega(\delta^{-2})$ samples to estimate $\mathbb{E}[Y_n^{(\pi)}]$ within error $O(\delta)$. Therefore, this naive (unimplementable) solution would require $N \times 2^p \times \delta^{-2}$ observations to estimate $\mathbb{E}[\mathbf{Y}_N^{(\Pi)}]$.

On the other hand, Synthetic Combinations produces consistent estimates of the potential outcome despite being given *at most only a single noisy sample* of each potential outcome. As the discussion after Theorem 6.6 shows, if $|\mathcal{I}| \leq |\Pi_n|$, Synthetic Combinations requires $|\mathcal{I}| \times r^4 s^2 p/\delta^2$ observations for the donor set, and $(N - |\mathcal{I}|) \times r^4/\delta^4$ observations for the non-donor units to achieve an estimation error of $O(\delta)$ for all $N \times 2^p$ causal parameters. Hence, we have that the number of observations required to achieve an estimation error of $O(\delta)$ for all pairs $(n, \pi)$ scales as $O\left(poly(r/\delta) \times (N + s^2 p)\right)$.

**Sample Complexity Comparison to Other Methods.** *Horizontal regression:* An alternative algorithm would be to learn an individual model for each unit $n \in [N]$. That is, run a separate horizontal regression via the Lasso for every unit. This alternative algorithm has sample complexity that scales at least as $O(N \times s^2 p/\delta^2)$ rather than $O\left(poly(r)/\delta^4 \times (N + s^2 p)\right)$ required by Synthetic Combinations. It suffers because it does not utilize any structure across units (i.e., the low-rank property of $\mathcal{A}$), whereas Synthetic Combinations captures the similarity between units via PCR.

*Matrix completion:* Synthetic Combinations can be thought of as a matrix completion method; estimating $\mathbb{E}[Y_n^{(\pi)}]$ is equivalent to imputing $(n, \pi)$-th entry of the observation matrix $\mathbf{Y} \in \{\mathbb{R} \cup \star\}^{N \times 2^p}$, where recall $\star$ denotes an unobserved unit-combination outcome. Under the low-rank property (Assumption 3.1(b)) and various models of missingness (i.e., observation patterns), recent works on matrix completion [14, 29, 6] (see Section 2 for an overview) have established that estimating $\mathbb{E}[Y_n^{(\pi)}]$ to an accuracy $O(\delta)$ requires at least $O(\text{poly}(r/\delta) \times (N + 2^p))$ samples. This is because matrix completion techniques do not leverage the sparsity of $\boldsymbol{\alpha}_n$. Moreover, matrix completion results typically report error in the Frobenius norm, whereas we give entry-wise guarantees. This leads to an extra factor of $s$ in our analysis as it requires proving convergence for $\|\hat{\boldsymbol{\alpha}}_n - \boldsymbol{\alpha}_n\|_1$ rather than $\|\hat{\boldsymbol{\alpha}}_n - \boldsymbol{\alpha}_n\|_2$. Hence, Synthetic Combinations combines the best of both approaches by leveraging both the structure of the potential outcomes *and* the similarity across units.

**Natural Lower Bound on Sample Complexity.** We provide an informal discussion on the lower bound sample-complexity to estimate all $N \times 2^p$ potential outcomes. As established in Lemma E.1, $\mathcal{A}$ has at most $rs$ non-zero columns. Counting the parameters in the singular value decomposition of $\mathcal{A}$, only $r \times (N + rs)$ free parameters are required to be estimated. Hence, a natural lower bound on the sample complexity scales as $O(Nr + r^2 s)$. Hence, Synthetic Combinations is only sub-optimal by a factor (ignoring logarithmic factors) of $sp$ and $\text{poly}(r)$. As discussed earlier, an additional factor of $s$ can be removed if we focus on deriving Frobenius norm error bounds. It remains as interesting future work to derive estimation procedures that are able to achieve this lower bound.

# 7   Conclusion

In this work, we formulate a causal inference framework for combinatorial interventions, a setting that is ubiquitous in practice. We propose a model that imposes both unit-specific structure, and latent similarity across units. Under this model, we propose an estimation procedure, Synthetic Combinations, that exploits the sparsity and low-rankness of the Fourier coefficients to efficiently estimate all $N \times 2^p$ causal parameters. We formally establish finite-sample consistency of Synthetic Combinations in an observational setting. Our work also naturally suggests future directions for research such as extending Synthetic Combinations to permutations over items (i.e., rankings), or providing an analysis of Synthetic Combinations that is agnostic to the horizontal regression algorithm used. A related line of work to the question above is also deriving estimation algorithms that can achieve the sample complexity lower bound discussed in Section 6.3.

# 8   Acknowledgements

We thank Alberto Abadie, Peng Ding, Giles Hooker, Devavrat Shah, Vasilis Syrgkanis, and Bin Yu for useful discussions and feedback.

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

# A  Combinatorial Inference Applications

In this section, we discuss how classical models and applications could relate to our proposed potential outcome model. In particular, we discuss two well-studied models for functions over combinations: low-degree Boolean polynomials and $k$-Juntas. We also discuss applications such as factorial design experiments and recommendation systems.

**Low-degree Boolean Polynomials.** A special instance of sparse Boolean functions is a low-degree polynomial which we define as follows. For a positive integer $d \leq p$, $f : \{-1,1\}^p \to \mathbb{R}$ is a $d$-degree polynomial if its Fourier transform satisfies the following for any input $\mathbf{x} \in \{-1,1\}^p$

$$f(\mathbf{x}) = \sum_{S \subset |p|, |S| \leq d} \alpha_S \chi_S(\mathbf{x}). \tag{6}$$

In this setting, we see that $s \leq \sum_{i=0}^{d} \binom{p}{i} \approx p^d$. That is, degree $d$-polynomials impose sparsity on the potential outcome by limiting the degree of interaction between interventions.

**Applications of low-degree Boolean polynomials.** We discuss two applications when the potential outcomes can be modeled as a low-degree Boolean polynomial.

*Factorial Design Experiments.* Factorial design experiments consist of $p$ treatments where each treatment arm can take on a discrete set of values, and units are assigned different combinations of these treatments. In the special case that each treatment arm only takes on two possible values, this experimental design mechanism is referred to as a $2^p$ factorial experiment. Factorial design experiments are widely employed in the social sciences, agricultural and industrial applications [20, 18, 38]. A common strategy to determine treatment effects in this setting is to assume that the potential outcome only depends on main effects and pairwise interactions with higher-order interactions being negligible [11, 21, 24]. That is, analysts implicitly enforce sparsity of $\boldsymbol{\alpha}_n$ by setting $d = 2$. We refer the reader to [39] for a detailed discussion of various estimation strategies and their validity in different settings. The model we propose (see Assumption 3.1) captures these various modeling choices commonly used in factorial design experiments by imposing sparsity on $\boldsymbol{\alpha}_n$. Further, as we show later, Synthetic Combinations can adapt to low-degree polynomials without pre-specifying the degree, $d$, i.e., it automatically adapts to the inherent level of interactions in the data. The additional assumption we make however is that there is structure across units, i.e., the matrix $\mathcal{A} = [\boldsymbol{\alpha}_n]_{n \in [N]}$ is low-rank.

*Optimal Team Selection.* Another use for combinatorial inference is to find the optimal configuration for a team. For example, a soccer team can be interested in finding the configuration of positions (i.e., defensive, midfield, offensive players) that leads to the highest winning percentage. Here, units $n$ are different teams, and combinations $\pi$ represent possible configurations of the team. The potential outcomes $\mathbb{E}[Y_n^{(\pi)}]$ represent the winning percentage for a given team $n$ under a particular configuration $\pi$, with the coefficients of $\boldsymbol{\alpha}_n$ representing the effect of different configurations on the win rate. In this setting, enforcing sparsity of $\boldsymbol{\alpha}_n$ via low-degree polynomials may be appropriate because the number of wins may depend on all the players, however only interactions between players of the same type matter while higher-order interactions between players of different types can be negligible. Returning to the example of a soccer team, the win rate can depend strongly on the synergy of the offensive players, whereas interactions between offensive and defensive players have a smaller effect. Further, since different teams might play in similar ways and can have the same optimal configuration, one potential way to capture this structure across teams is by placing a low-rank structure on $\mathcal{A}$.

$k$**-Juntas.** Another special case of sparse Boolean functions are $k$-Juntas which only depend on $k < p$ input variables. More formally, a function $f : \{-1,1\}^p \to \mathbb{R}$ is a $k$-junta if there exists a set $K \subset [p]$ with $|K| = k$ such that the fourier expansion of $f$ can be represented as follows

$$f(\mathbf{x}) = \sum_{S \subset K} \alpha_S \chi_S(\mathbf{x}) \tag{7}$$

Therefore, in the setting of the $k$-Junta, the sparsity index $s \leq 2^k$. In contrast to low-degree polynomials where the function depends on all $p$ variables but limits the degree of interaction between variables, $k$-Juntas only depend on $k$ variables but allow for arbitrary interactions amongst them.

**Applications of $k$-Juntas.** We discuss two applications when the potential outcomes can be modeled as a $k$-Junta.

*Recommendation Systems.* Recommendation platforms such as *Netflix* are often interested in recommending a combination of movies that maximizes a user's engagement with a platform. Here, units $n$ can be individual users, and $\pi$ represents combinations of different movies. The potential outcomes $\mathbb{E}[Y_n^{(\pi)}]$ are the engagement levels for a unit $n$ when presented with a combination of movies $\pi$, with $\boldsymbol{\alpha}_n$ representing the preferences for that user. In this setting, a user's engagement with the platform may only depend on a small subset of movies. For example, a user who is only interested in fantasy will only remain on the platform if they are recommended movies such as *Harry Potter*. Under this behavioral model, the potential outcomes (i.e., engagement levels) can be modeled as a $k$-Junta with the non-zero coefficients of $\boldsymbol{\alpha}_n$ representing the combinations of the $k$ movies that affect engagement level for a user $n$. Our potential outcome observational model (Assumption 3.1) captures this form of sparsity while also reflecting the low-rank structure commonly assumed when studying recommendation systems. Once again, our results show Synthetic Combinations can adapt to $k-$Juntas without pre-specifying the subset of features $K$, i.e., it automatically learns the important features for a given user.

*Knock-down Experiments in Genomics.* A key task in genomics is to identify which set of genes are responsible for a phenotype (i.e., physical trait of interest such as blood pressure) in a given individual. To do so, geneticists use knock-down experiments which measure the difference in the phenotype after eliminating the effect of a set of genes in an individual. To encode this process in the language of combinatorial causal inference, we can think of units $n$ as different individuals, an action as knocking a particular gene out, and $\pi$ as a combination of genes that are knocked out. The potential outcomes $\mathbb{E}[Y_n^{(\pi)}]$ is the expression of the phenotype for a unit $n$ when the combination of genes $\pi$ are eliminated via knock-down experiments, and the coefficients of $\boldsymbol{\alpha}_n$ represent the effect of different combination of genes on the phenotype. A typical assumption in genomics is that phenotypes only depend on a small set of genes and their interactions. In this setting, we can model this form of sparsity by thinking of the potential outcome function as a $k$-junta, as well as capturing the similarity between the effect of genes on different individuals via our low-rank assumption.

## B  Proof of Theorem 4.2

*Proof.* Below, the symbol $\overset{AX}{=}$ and $\overset{DX}{=}$ imply that the equality follows from Assumption $X$ and Definition $X$, respectively. We begin with the proof of Theorem 4.2 (a). *Proof of Theorem 4.2 (a):* For a donor unit $u \in \mathcal{I}$ and $\pi \in \Pi \backslash \Pi_u$, we have

$$
\begin{aligned}
\mathbb{E}[Y_u^{(\pi)} \mid \mathcal{A}] &\overset{A3.1}{=} \mathbb{E}[\langle \boldsymbol{\alpha}_u, \boldsymbol{\chi}^\pi \rangle + \epsilon_u^\pi \mid \mathcal{A}] \\
&\overset{A3.1(c)}{=} \langle \boldsymbol{\alpha}_u, \boldsymbol{\chi}^\pi \rangle \mid \mathcal{A} \\
&= \langle \boldsymbol{\alpha}_u, \boldsymbol{\chi}^\pi \rangle \mid \mathcal{A}, \mathcal{D} \\
&\overset{A3.1(b)}{=} \langle \boldsymbol{\alpha}_u, \tilde{\boldsymbol{\chi}}_u^\pi \rangle \mid \mathcal{A}, \mathcal{D} \\
&\overset{A4.1(a)}{=} \langle \boldsymbol{\alpha}_u, \sum_{\pi_u \in \Pi_u} \beta_{\pi_u}^\pi \tilde{\boldsymbol{\chi}}_u^{\pi_u} \rangle \mid \mathcal{A}, \mathcal{D} \\
&= \sum_{\pi_u \in \Pi_u} \beta_{\pi_u}^\pi \langle \boldsymbol{\alpha}_u, \tilde{\boldsymbol{\chi}}_u^{\pi_u} \rangle \mid \mathcal{A}, \mathcal{D} \\
&\overset{A3.1(c), A3.2}{=} \sum_{\pi_u \in \Pi_u} \beta_{\pi_u}^\pi \mathbb{E}[\langle \boldsymbol{\alpha}_u, \tilde{\boldsymbol{\chi}}_u^{\pi_u} \rangle + \epsilon_u^{\pi_u} \mid \mathcal{A}, \mathcal{D}] \\
&\overset{A3.1}{=} \sum_{\pi_u \in \Pi_u} \beta_{\pi_u}^\pi \mathbb{E}[Y_{u, \pi_u} \mid \mathcal{A}, \mathcal{D}]
\end{aligned}
$$

*Proof of Theorem 4.2 (b):* For a donor unit $n \in [N] \setminus I^D$ and $\pi \in \Pi \setminus \Pi_n$, we have

$$
\begin{aligned}
\mathbb{E}[Y_n^{(\pi)} \mid \mathcal{A}] &\overset{A3.1}{=} \mathbb{E}[\langle \boldsymbol{\alpha}_n, \boldsymbol{\chi}^\pi \rangle + \epsilon_n^\pi \mid \mathcal{A}] \\
&\overset{A3.1(c)}{=} \langle \boldsymbol{\alpha}_n, \boldsymbol{\chi}^\pi \rangle \mid \mathcal{A} \\
&= \langle \boldsymbol{\alpha}_n, \boldsymbol{\chi}^\pi \rangle \mid \mathcal{A}, \mathcal{D} \\
&\overset{A4.1(b)}{=} \langle \sum_{u \in \mathcal{I}} w_u^n \boldsymbol{\alpha}_u, \boldsymbol{\chi}^\pi \rangle \mid \mathcal{A}, \mathcal{D} \\
&= \sum_{u \in \mathcal{I}} w_u^n \langle \boldsymbol{\alpha}_u, \boldsymbol{\chi}^\pi \rangle \mid \mathcal{A}, \mathcal{D} \\
&\overset{A3.1(c), A3.2}{=} \sum_{u \in \mathcal{I}} w_u^n \mathbb{E}[\langle \boldsymbol{\alpha}_u, \boldsymbol{\chi}^\pi \rangle + \epsilon_u^\pi \mid \mathcal{A}, \mathcal{D}] \\
&\overset{A3.1}{=} \sum_{u \in \mathcal{I}} w_u^n \mathbb{E}[Y_u^{(\pi)} \mid \mathcal{A}, \mathcal{D}] \\
&= \sum_{u \in \mathcal{I}} \sum_{\pi_u \in \Pi_u} w_u^n \beta_{\pi_u}^\pi \mathbb{E}[Y_{u, \pi_u} \mid \mathcal{A}, \mathcal{D}]
\end{aligned}
$$

where the last equality follows from Theorem 4.2 (a). $\qquad \square$

## C    Proof of Theorem 6.6

### C.1    Proof of Theorem 6.6 (a)

We have that

$$
\begin{aligned}
\hat{\mathbb{E}}[Y_u^{(\pi)}] - \mathbb{E}[Y_u^{(\pi)}] &= \langle \hat{\boldsymbol{\alpha}}^u, \boldsymbol{\chi}^\pi \rangle - \langle \boldsymbol{\alpha}_u, \boldsymbol{\chi}^\pi \rangle \\
&= \langle \hat{\boldsymbol{\alpha}}_u - \boldsymbol{\alpha}_u, \boldsymbol{\chi}^\pi \rangle \\
&\leq \|\hat{\boldsymbol{\alpha}}_u - \boldsymbol{\alpha}_u\|_1 \|\boldsymbol{\chi}^\pi\|_\infty \\
&= \|\hat{\boldsymbol{\alpha}}_u - \boldsymbol{\alpha}_u\|_1
\end{aligned}
\tag{8}
$$

To finish the proof, we quote the following Theorem which we adapt to our notation.

**Theorem C.1** (Theorem 2.18 in [36])**.** *Fix the number of samples $n \geq 2$. Assume that the linear model $Y = \mathbf{X}\theta^* + \epsilon$, where $\mathbf{X} \in \mathbb{R}^{n \times d}$ and $\epsilon$ is a sub-gaussian random variable with noise variance $\sigma^2$. Moreover, assume that $\|\theta^*\|_0 \leq k$, and that $\mathbf{X}$ satisfies the incoherence condition (Assumption 6.3) with parameter $k$. Then, the lasso estimator $\hat{\theta}^L$ with regularization parameter defined by*

$$
2\tau = 8\sigma \sqrt{\log(2d)/n} + 8\sigma \sqrt{\log(1/\delta)/n}
$$

*satisfies*

$$
\|\theta^* - \hat{\theta}^L\|_2^2 \leq k\sigma^2 \frac{\log(2d/\delta)}{n}
\tag{9}
$$

*with probability at least $1 - \delta$.*

Further, as in established in the proof of Theorem 2.18 in [36], $\|\theta^* - \hat{\theta}^L\|_1 \leq \sqrt{k}\|\theta^* - \hat{\theta}^L\|_2$. Note that the set-up of Theorem C.1 holds in our setting with the following notational changes: $Y = \mathbf{Y}_{\Pi_u}$, $\mathbf{X} = \boldsymbol{\chi}(\Pi_u)$, $\theta^* = \boldsymbol{\alpha}_u$, $\hat{\theta}^L = \hat{\boldsymbol{\alpha}}_u$, $k = s$ as well as our assumptions on the regularization parameter $\lambda_u$ and that $\epsilon_u^\pi$ is sub-gaussian (Assumption 6.2). Applying Theorem C.1 gives us

$$
\|\hat{\boldsymbol{\alpha}}_u - \boldsymbol{\alpha}_u\|_1 = O_p\left( \sqrt{\frac{s^2 p}{|\Pi_u|}} \right)
$$

Substituting this bound into (8) yields the claimed result.

## C.2 Proof of Theorem 6.6 (b)

For any matrix $\mathbf{A}$ with orthonormal columns, let $\mathcal{P}_A = \mathbf{A}\mathbf{A}^T$ denote the projection matrix on the subspace spanned by the columns of $\mathbf{A}$. Define $\tilde{\mathbf{w}}^n = \mathcal{P}_{V_{\mathcal{I}}^{(\Pi_n)}}\mathbf{w}^n$, where $\mathbf{V}_{\mathcal{I}}^{(\Pi_n)}$ are the right singular vectors of $\mathbb{E}[\mathbf{Y}_{\mathcal{I}}^{(\Pi_n)}]$. Let $\Delta_w^n = \hat{\mathbf{w}}^n - \tilde{\mathbf{w}}^n \in \mathbb{R}^{|\mathcal{I}|}$, and $\Delta_{\mathcal{I}}^{\pi} = \hat{\mathbb{E}}[\mathbf{Y}_{\mathcal{I}}^{(\pi)}] - \mathbb{E}[\mathbf{Y}_{\mathcal{I}}^{(\pi)}] \in \mathbb{R}^{|\mathcal{I}|}$. Denote $\Delta_E = \max_{u \in \mathcal{I}, \pi \in \Pi} \left| \hat{\mathbb{E}}[Y_u^{(\pi)}] - \mathbb{E}[Y_u^{(\pi)}] \right|$. In order to proceed, we first state the following result,

**Lemma C.2.** *Let the set-up of Theorem 6.6 hold. Then, we have*

$$\mathbb{E}[Y_n^{(\pi)}] = \langle \mathbb{E}[\mathbf{Y}_{\mathcal{I}}^{(\pi)}], \tilde{\mathbf{w}}^n \rangle$$

Using Lemma C.2, and the notation established above, we have

$$\left| \hat{\mathbb{E}}[Y_n^{(\pi)}] - \mathbb{E}[Y_n^{(\pi)}] \right| = \left| \langle \hat{\mathbb{E}}[\mathbf{Y}_{\mathcal{I}}^{(\pi)}], \hat{\mathbf{w}}^n \rangle - \langle \mathbb{E}[\mathbf{Y}_{\mathcal{I}}^{(\pi)}], \tilde{\mathbf{w}}^n \rangle \right|$$

$$\leq |\langle \Delta_{\mathcal{I}}^{\pi}, \tilde{\mathbf{w}}^n \rangle| + \left| \langle \mathbb{E}[\mathbf{Y}_{\mathcal{I}}^{(\pi)}], \Delta_w^n \rangle \right| + |\langle \Delta_w^n, \Delta_{\mathcal{I}}^{\pi} \rangle| \tag{10}$$

From Assumption 6.5, it follows that $\mathbb{E}[\mathbf{Y}_{\mathcal{I}}^{(\pi)}] = \mathcal{P}_{V_{\mathcal{I}}^{(\Pi_n)}}\mathbb{E}[\mathbf{Y}_{\mathcal{I}}^{(\pi)}]$, where $\mathbf{V}_{\mathcal{I}}^{(\Pi_n)}$ are the right singular vectors of $\mathbb{E}[\mathbf{Y}_{\mathcal{I}}^{(\Pi_n)}]$. Using this in (10) gives us

$$\left| \hat{\mathbb{E}}[Y_n^{(\pi)}] - \mathbb{E}[Y_n^{(\pi)}] \right| \leq |\langle \Delta_{\mathcal{I}}^{\pi}, \tilde{\mathbf{w}}^n \rangle| + \left| \langle \mathbb{E}[\mathbf{Y}_{\mathcal{I}}^{(\pi)}], \mathcal{P}_{V_{\mathcal{I}}^{(\Pi_n)}}\Delta_w^n \rangle \right| + |\langle \Delta_w^n, \Delta_{\mathcal{I}}^{\pi} \rangle| \tag{11}$$

Below, we bound the three terms on the right-hand-side of (11) separately. Before we bound each term, we state a Lemma that will be useful to help us establish the results.

**Lemma C.3.** *Let Assumptions 3.1, 4.1, 6.1, and 6.4 hold. Then,* $\|\tilde{\mathbf{w}}^n\|_2 \lesssim \sqrt{\frac{r_n}{|\mathcal{I}|}}$

*Bounding Term 1.* By Holder's inequality and Lemma C.3, we have that

$$|\langle \Delta_{\mathcal{I}}^{\pi}, \tilde{\mathbf{w}}^n \rangle| \leq \|\tilde{\mathbf{w}}^n\|_1 \|\Delta_{\mathcal{I}}^{\pi}\|_{\infty} \leq \|\tilde{\mathbf{w}}^n\|_1 \Delta_E \leq \sqrt{|\mathcal{I}|}\|\tilde{\mathbf{w}}^n\|_2 \Delta_E \lesssim \sqrt{r_n}\Delta_E \tag{12}$$

This concludes the analysis for the first term.

*Bounding Term 2.* By Cauchy-Schwarz and Assumption 6.1 we have,

$$\left| \langle \mathbb{E}[\mathbf{Y}_{\mathcal{I}}^{(\pi)}], \mathcal{P}_{V_{\mathcal{I}}^{(\Pi_n)}}\Delta_w^n \rangle \right| \leq \|\mathbb{E}[\mathbf{Y}_{\mathcal{I}}^{(\pi)}]\|_2 \|\mathcal{P}_{V_{\mathcal{I}}^{(\Pi_n)}}\Delta_w^n\|_2$$

$$\leq \sqrt{|\mathcal{I}|}\|\mathcal{P}_{V_{\mathcal{I}}^{(\Pi_n)}}\Delta_w^n\|_2 \tag{13}$$

We now state a lemma that will help us conclude our bound of Term 2. The proof is given in Appendix D.3.

**Lemma C.4.** *Let the set-up of Theorem 6.6 hold. Then,*

$$\|\mathcal{P}_{V_{\mathcal{I}}^{(\Pi_n)}}\Delta_w^n\|_2$$

$$= \tilde{O}_p\left( r_n^2 \left[ \frac{\Delta_E}{\sqrt{|\mathcal{I}|}\min\{\sqrt{|\Pi_n|}, \sqrt{|\mathcal{I}|}\}} + \Delta_E^2 \right] + \frac{r_n^{3/2}\Delta_E}{\sqrt{|\mathcal{I}|}} + \frac{r_n\left(1 + \sqrt{\Delta_E}r_n^{1/4}\right)}{\sqrt{|\mathcal{I}|}|\Pi_n|^{1/4}} \right) \tag{14}$$

Incorporating the result of the lemma above into (13) gives us

$$\left| \langle \mathbb{E}[\mathbf{Y}_{\mathcal{I}}^{(\pi)}], \mathcal{P}_{V_{\mathcal{I}}^{(\Pi_n)}}\Delta_w^n \rangle \right|$$

$$= \tilde{O}_p\left( r_n^2 \left[ \frac{\Delta_E}{\min\{\sqrt{|\Pi_n|}, \sqrt{|\mathcal{I}|}\}} + \sqrt{|\mathcal{I}|}\Delta_E^2 \right] + r_n^{3/2}\Delta_E + \frac{r_n\left(1 + \sqrt{\Delta_E}r_n^{1/4}\right)}{|\Pi_n|^{1/4}} \right) \tag{15}$$

*Bounding Term 3.* By Holder's inequality, we have that

$$|\langle \Delta_w^n, \Delta_{\mathcal{I}}^\pi \rangle| \le \|\Delta_w^n\|_2 \|\Delta_{\mathcal{I}}^\pi\|_2$$
$$\le \sqrt{|\mathcal{I}|}\|\Delta_w^n\|_2 \|\Delta_{\mathcal{I}}^\pi\|_\infty$$
$$\le \sqrt{|\mathcal{I}|}\Delta_E\|\Delta_w^n\|_2 \tag{16}$$
$$\tag{17}$$

We now state a proposition that will help us conclude our proof of Term 3. The proof is given in Appendix D.4.

**Proposition C.5.** *Let the set-up of Theorem 6.6 hold. Then, conditioned on $\mathcal{A}$, we have*

$$\hat{\mathbf{w}}^n - \tilde{\mathbf{w}}^n = \tilde{O}_p\left( r_n \left[ \frac{\|\tilde{\mathbf{w}}^n\|_2}{\min\{\sqrt{|\Pi_n|}, \sqrt{|\mathcal{I}|}\}} + \Delta_E \right] \right) \tag{18}$$

As a result of Proposition C.5 and Lemma C.3, we have the following bound for Term 3,

$$|\langle \Delta_w^n, \Delta_{\mathcal{I}}^\pi \rangle| = \tilde{O}_p\left( r_n^{3/2} \left[ \frac{\Delta_E}{\min\{\sqrt{|\Pi_n|}, \sqrt{|\mathcal{I}|}\}} + \sqrt{|\mathcal{I}|}\Delta_E^2 \right] \right) \tag{19}$$

*Collecting Terms.* Combining equations (12), (15), (19) gives us

$$\left| \hat{\mathbb{E}}[Y_n^{(\pi)}] - \mathbb{E}[Y_n^{(\pi)}] \right|$$
$$= \tilde{O}_p\left( r_n^2 \left[ \frac{\Delta_E}{\min\{\sqrt{|\Pi_n|}, \sqrt{|\mathcal{I}|}\}} + \sqrt{|\mathcal{I}|}\Delta_E^2 \right] + r_n^{3/2}\Delta_E + \frac{r_n\left(1 + \sqrt{\Delta_E}r_n^{1/4}\right)}{|\Pi_n|^{1/4}} \right) \tag{20}$$

By Theorem 6.6 (a), we have that

$$\Delta_E = \max_{u \in \mathcal{I}} O_p\left( \sqrt{\frac{s^2 p}{|\Pi_u|}} \right) = O_p\left( \sqrt{\frac{s^2 p}{M}} \right) \tag{21}$$

where we remind the reader that $M = \min_{u \in \mathcal{I}} |\Pi_u|$. Substituting (21), and our assumption that $M = \omega(r_n^2 s^2 p)$ into (20), we get

$$\left| \hat{\mathbb{E}}[Y_n^{(\pi)}] - \mathbb{E}[Y_n^{(\pi)}] \right| = \tilde{O}_p\left( \frac{r_n^2 \sqrt{s^2 p}}{\sqrt{M \times \min\{|\Pi_n|, |\mathcal{I}|\}}} + \frac{r_n^2 s^2 p \sqrt{|\mathcal{I}|}}{M} + \frac{r_n}{|\Pi_n|^{1/4}} \right) \tag{22}$$

# D   Proofs of Helper Lemmas for Theorem 6.6.

In this section we provide proofs of Lemmas C.2, C.3, C.4, and Proposition C.5 which were required for the Proof of Theorem 6.6.

## D.1   Proof of Lemma C.2

By Assumption 3.1 and 4.1 (b), we have

$$\mathbb{E}[Y_n^{(\pi)}] = \mathbb{E}[\langle \boldsymbol{\alpha}_n, \boldsymbol{\chi}^\pi \rangle + \epsilon_n^\pi]$$
$$= \mathbb{E}[\langle \boldsymbol{\alpha}_n, \boldsymbol{\chi}^\pi \rangle]$$
$$= \sum_{u \in \mathcal{I}} w_u^n \mathbb{E}[\langle \boldsymbol{\alpha}_u, \boldsymbol{\chi}^\pi \rangle]$$
$$= \sum_{u \in \mathcal{I}} w_u^n \mathbb{E}[Y_u^{(\pi)}]$$
$$= \mathbb{E}[\mathbf{Y}_{\mathcal{I}}^{(\pi)}]^T \mathbf{w}^n$$

From Assumption 6.5, we have that $\mathbb{E}[\mathbf{Y}_{\mathcal{I}}^{(\pi)}] = \mathcal{P}_{V_{\mathcal{I}}^{(\Pi_n)}}\mathbb{E}[\mathbf{Y}_{\mathcal{I}}^{(\pi)}]$. Substituting this into the equation above completes the proof.

## D.2 Proof of Lemma C.3

For simplicity, denote $\mathbb{E}[\mathbf{Y}_n^{(\Pi_n)}] = \mathbb{E}[\mathbf{Y}^{(\Pi_n)}]$. By definition, $\tilde{\mathbf{w}}^n$ is the solution to the following optimization program

$$\min_{\mathbf{w} \in \mathbb{R}^{|\mathcal{I}|}} \quad \|\mathbf{w}\|_2$$
$$\text{s.t.} \quad \mathbb{E}[\mathbf{Y}^{(\Pi_n)}] = \mathbb{E}[\mathbf{Y}_{\mathcal{I}}^{(\Pi_n)}]\mathbf{w} \tag{23}$$

Let $\mathbf{U}_{\mathcal{I}}^{(\Pi_n)}$, $\boldsymbol{\Sigma}_{\mathcal{I}}^{(\Pi_n)}$, $\mathbf{V}_{\mathcal{I}}^{(\Pi_n)}$ denote the SVD of $\mathbb{E}[\mathbf{Y}_{\mathcal{I}}^{(\Pi_n)}]$. Further, let $\mathbf{U}_{\mathcal{I}}^{(\Pi_n),r_n}$, $\boldsymbol{\Sigma}_{\mathcal{I}}^{(\Pi_n),r_n}$, $\mathbf{V}_{\mathcal{I}}^{(\Pi_n),r_n}$ denote the rank $r_n$ truncation of the SVD. Then, define $\mathbf{w}_{r_n} = \mathbf{V}_{\mathcal{I}}^{(\Pi_n),r_n}(\boldsymbol{\Sigma}_{\mathcal{I}}^{(\Pi_n),r_n})^\dagger(\mathbf{U}_{\mathcal{I}}^{(\Pi_n),r_n})^T\mathbb{E}[\mathbf{Y}^{(\Pi_n)}]$, where $\dagger$ is pseudo-inverse. We first show that $\mathbf{w}_{r_n}$ is a solution to (23).

$$\mathbb{E}[\mathbf{Y}_{\mathcal{I}}^{(\Pi_n)}]\mathbf{w}_{r_n} = \left(\mathbf{U}_{\mathcal{I}}^{(\Pi_n)}\boldsymbol{\Sigma}_{\mathcal{I}}^{(\Pi_n)}(\mathbf{V}_{\mathcal{I}}^{(\Pi_n)})^T\right)\mathbf{V}_{\mathcal{I}}^{(\Pi_n),r_n}(\boldsymbol{\Sigma}_{\mathcal{I}}^{(\Pi_n),r_n})^\dagger(\mathbf{U}_{\mathcal{I}}^{(\Pi_n),r_n})^T\mathbb{E}[\mathbf{Y}^{(\Pi_n)}]$$

$$= \left(\sum_{i=1}^{r_n} s_i u_i v_i^T\right)\left(\sum_{j=1}^{r_n} \frac{1}{s_j} v_j u_j^T \mathbb{E}[\mathbf{Y}^{(\Pi_n)}]\right)$$

$$= \sum_{i,j=1}^{r_n} \frac{s_i}{s_j} u_i v_i^T v_j u_j^T \mathbb{E}[\mathbf{Y}^{(\Pi_n)}]$$

$$= \sum_{i=1}^{r_n} u_i u_i^T \mathbb{E}[\mathbf{Y}^{(\Pi_n)}]$$

$$= \mathbb{E}[\mathbf{Y}^{(\Pi_n)}]$$

Next, we bound $\|\mathbf{w}_{r_n}\|_2$ using Assumptions 6.1 and 6.4 as follows

$$\|\mathbf{w}_{r_n}\|_2 \leq \|(\boldsymbol{\Sigma}_{\mathcal{I}}^{(\Pi_n),r_n})^\dagger\|_2\|\mathbb{E}[\mathbf{Y}^{(\Pi_n)}]\|_2$$

$$\leq \frac{\sqrt{|\Pi_n|}}{s_{r_n}(\mathbb{E}[\mathbf{Y}_{\mathcal{I}}^{(\Pi_n)}])}$$

$$\leq \sqrt{\frac{cr_n}{|\mathcal{I}|}}$$

Therefore, we have

$$\|\tilde{\mathbf{w}}^n\|_1 \leq \sqrt{|\mathcal{I}|}\|\tilde{\mathbf{w}}_{r_n}\|_2 \leq \sqrt{cr_n}$$

## D.3 Proof of Lemma C.4

First, we introduce some necessary notation required for the proof. Let $\hat{\mathbb{E}}\left[\mathbf{Y}_{\mathcal{I}}^{(\Pi_n)}\right] = \hat{\mathbf{U}}_{\mathcal{I}}^{(\Pi_n)}\hat{\mathbf{S}}_{\mathcal{I}}^{(\Pi_n)}\hat{\mathbf{V}}_{\mathcal{I}}^{(\Pi_n)}$ denote the rank $r_n$ SVD of $\hat{\mathbb{E}}\left[\mathbf{Y}_{\mathcal{I}}^{(\Pi_n)}\right]$. Then, to establish Lemma C.4, consider the following decomposition:

$$\mathcal{P}_{V_{\mathcal{I}}^{(\Pi_n)}}\Delta_w^n = \left(\mathcal{P}_{V_{\mathcal{I}}^{(\Pi_n)}} - \mathcal{P}_{\hat{V}_{\mathcal{I}}^{(\Pi_n)}}\right)\Delta_w^n + \mathcal{P}_{\hat{V}_{\mathcal{I}}^{(\Pi_n)}}\Delta_w^n$$

We bound each of these terms separately again.

*Bounding Term 1.* We have

$$\|\left(\mathcal{P}_{V_{\mathcal{I}}^{(\Pi_n)}} - \mathcal{P}_{\hat{V}_{\mathcal{I}}^{(\Pi_n)}}\right)\Delta_w^n\|_2 \leq \left\|\mathcal{P}_{V_{\mathcal{I}}^{(\Pi_n)}} - \mathcal{P}_{\hat{V}_{\mathcal{I}}^{(\Pi_n)}}\right\|_{\text{op}}\|\Delta_w^n\|_2 \tag{24}$$

**Theorem D.1** (Wedin's Theorem [37]). *Given $\mathbf{A}, \mathbf{B} \in \mathbb{R}^{m\times n}$, let $(\mathbf{U}, \mathbf{V}), (\hat{\mathbf{U}}, \hat{\mathbf{V}})$ denote their respective left and right singular vectors. Further, let $(\mathbf{U}_k, \mathbf{V}_k) \in$(respectively, $(\hat{\mathbf{U}}_k, \hat{\mathbf{V}}_k)$) correspond to the truncation of $(\mathbf{U}, \mathbf{V})$ (respectively, $(\hat{\mathbf{U}}, \hat{\mathbf{V}})$), respectively, that retains the columns correspondng to the top $k$ singular values of $\mathbf{A}$ (respectively, $\mathbf{B}$). Let $s_i$ represent the $i$-th singular values of A. Then,*
$$\max(\|\mathcal{P}_{U_k} - \mathcal{P}_{\hat{U}_k}\|_{op}, \|\mathcal{P}_{V_k} - \mathcal{P}_{\hat{V}_k}\|_{op}) \leq \frac{2\|\mathbf{A}-\mathbf{B}\|_{op}}{s_k - s_{k+1}}$$

Applying Theorem D.1 gives us

$$\max\left(\left\|\mathcal{P}_{U_{\mathcal{I}}^{(\Pi_n)}}-\mathcal{P}_{\hat{U}_{\mathcal{I}}^{(\Pi_n)}}\right\|_{\text{op}},\left\|\mathcal{P}_{V_{\mathcal{I}}^{(\Pi_n)}}-\mathcal{P}_{\hat{V}_{\mathcal{I}}^{(\Pi_n)}}\right\|_{\text{op}}\right)\leq\frac{2\|\mathbb{E}[\mathbf{Y}_{\mathcal{I}}^{(\Pi_n)}]-\hat{\mathbb{E}}[\mathbf{Y}_{\mathcal{I}}^{(\Pi_n)}]\|_{\text{op}}}{s_{r_n}-s_{r_n+1}}$$

$$\leq\frac{2\sqrt{|\mathcal{I}||\Pi_n|}\|\mathbb{E}[\mathbf{Y}_{\mathcal{I}}^{(\Pi_n)}]-\hat{\mathbb{E}}[\mathbf{Y}_{\mathcal{I}}^{(\Pi_n)}]\|_{\text{max}}}{s_{r_n}-s_{r_n+1}}$$

$$=\frac{2\sqrt{|\mathcal{I}||\Pi_n|}\Delta_E}{s_{r_n}-s_{r_n+1}}$$

$$=\frac{2\sqrt{|\mathcal{I}||\Pi_n|}\Delta_E}{s_{r_n}}\tag{25}$$

where the last equality follows from the fact that $\text{rank}(\mathbb{E}[\mathbf{Y}_{\mathcal{I}}^{\Pi_n}])=r_n$, hence $s_{r_n+1}=0$. Now, plugging Assumption 6.4, (25) into (24) gives us

$$\max\left(\left\|\mathcal{P}_{V_{\mathcal{I}}^{(\Pi_n)}}-\mathcal{P}_{\hat{V}_{\mathcal{I}}^{(\Pi_n)}}\right\|_{\text{op}},\left\|\mathcal{P}_{V_{\mathcal{I}}^{(\Pi_n)}}-\mathcal{P}_{\hat{V}_{\mathcal{I}}^{(\Pi_n)}}\right\|_{\text{op}}\right)\leq C\sqrt{r_n}\Delta_E\tag{26}$$

Substituting the result of Proposition C.5 and (26) into (24) gives us

$$\left\|\left(\mathcal{P}_{V_{\mathcal{I}}^{(\Pi_n)}}-\mathcal{P}_{\hat{V}_{\mathcal{I}}^{(\Pi_n)}}\right)\Delta_w^n\right\|_2=O_p\left(\log^3(|\Pi_n||\mathcal{I}|)r_n^{3/2}\left[\frac{\|\tilde{\mathbf{w}}^n\|_2\Delta_E}{\min\{\sqrt{|\Pi_n|},\sqrt{|\mathcal{I}|}\}}+\Delta_E^2\right]\right)\tag{27}$$

To further simplify (27), we substitute the result of Lemma C.3

Substituting the Lemma C.3 into (27) gives us

$$\left\|\left(\mathcal{P}_{V_{\mathcal{I}}^{(\Pi_n)}}-\mathcal{P}_{\hat{V}_{\mathcal{I}}^{(\Pi_n)}}\right)\Delta_w^n\right\|_2=\tilde{O}_p\left(r_n^2\left[\frac{\Delta_E}{\sqrt{|\mathcal{I}|}\min\{\sqrt{|\Pi_n|},\sqrt{|\mathcal{I}|}\}}+\Delta_E^2\right]\right)\tag{28}$$

*Bounding Term 2.* We introduce some necessary notation required to bound the second term. Let $\hat{\mathbb{E}}[\mathbf{Y}_{\mathcal{I}}^{(\Pi_n),r_n}]=\sum_{l=1}^{r_n}\hat{s}_l\hat{\mu}_l\hat{v}_l'$ denote the $r_n$ decomposition of $\hat{\mathbb{E}}[\mathbf{Y}_{\mathcal{I}}^{(\Pi_n)}]$. Let, $\hat{\mathbb{E}}[\mathbf{Y}_{\mathcal{I}}^{(\Pi_n),r_n}]=\hat{\mathbf{U}}_{\mathcal{I}}^{(\Pi_n)}\hat{\mathbf{S}}_{\mathcal{I}}^{(\Pi_n)}(\hat{\mathbf{V}}_{\mathcal{I}}^{(\Pi_n)})^T$. Further, define $\epsilon^{\Pi_n}=[\epsilon_n^\pi:\pi\in\Pi_n]\in\mathbb{R}^{|\Pi_n|}$.
Then to begin, note that since $\hat{\mathbf{V}}_{\mathcal{I}}^{(\Pi_n)}$ is a isometry, we have that

$$\|\mathcal{P}_{\hat{V}_{\mathcal{I}}^{(\Pi_n)}}\Delta_w^n\|_2^2=\|(\hat{\mathbf{V}}_{\mathcal{I}}^{(\Pi_n)})^T\Delta_w^n\|_2^2$$

To upper bound $\|(\hat{\mathbf{V}}_{\mathcal{I}}^{(\Pi_n)})^T\Delta_w^n\|_2^2$ as follows, consider

$$\|\hat{\mathbb{E}}[\mathbf{Y}_{\mathcal{I}}^{(\Pi_n)}]\Delta_w^n\|_2^2=\left((\hat{\mathbf{V}}_{\mathcal{I}}^{(\Pi_n)})^T\Delta_w^n\right)^T\left(\hat{\mathbf{S}}_{\mathcal{I}}^{(\Pi_n)}\right)^2\left((\hat{\mathbf{V}}_{\mathcal{I}}^{(\Pi_n)})^T\Delta_w^n\right)\geq\hat{s}_{r_n}^2\|(\hat{\mathbf{V}}_{\mathcal{I}}^{(\Pi_n)})^T\Delta_w^n\|_2^2$$

Using the two equations above gives us

$$\|\mathcal{P}_{\hat{V}_{\mathcal{I}}^{(\Pi_n)}}\Delta_w^n\|_2^2\leq\frac{\|\hat{\mathbb{E}}[\mathbf{Y}_{\mathcal{I}}^{(\Pi_n),r_n}]\Delta_w^n\|_2^2}{\hat{s}_{r_n}^2}\tag{29}$$

To bound the numerator in (29), note that by definition $\mathbb{E}[\mathbf{Y}^{(\Pi_n)}]=\mathbb{E}[\mathbf{Y}_{\mathcal{I}}^{(\Pi_n)}]\tilde{\mathbf{w}}^n$. Using this observation, we have

$$\|\hat{\mathbb{E}}[\mathbf{Y}_{\mathcal{I}}^{(\Pi_n),r_n}]\Delta_w^n\|_2^2\leq2\|\hat{\mathbb{E}}[\mathbf{Y}_{\mathcal{I}}^{(\Pi_n),r_n}]\hat{\mathbf{w}}^n-\mathbb{E}[\mathbf{Y}^{(\Pi_n)}]\|_2^2+2\|\hat{\mathbb{E}}[\mathbf{Y}_{\mathcal{I}}^{(\Pi_n),r_n}]\tilde{\mathbf{w}}^n-\mathbb{E}[\mathbf{Y}^{(\Pi_n)}]\|_2^2$$

$$=2\|\hat{\mathbb{E}}[\mathbf{Y}_{\mathcal{I}}^{(\Pi_n),r_n}]\hat{\mathbf{w}}^n-\mathbb{E}[\mathbf{Y}^{(\Pi_n)}]\|_2^2+2\|(\hat{\mathbb{E}}[\mathbf{Y}_{\mathcal{I}}^{(\Pi_n),r_n}]-\mathbb{E}[\mathbf{Y}_{\mathcal{I}}^{(\Pi_n)}])\tilde{\mathbf{w}}^n\|_2^2\tag{30}$$

To proceed, we then use the following inequality: for any $\mathbf{A}\in\mathbb{R}^{a\times b},v\in\mathbb{R}^b$, we have

$$\|\mathbf{A}v\|_2=\|\sum_{j=1}^b\mathbf{A}_{.j}v_j\|_2\leq\left(\max_{j\leq b}\|\mathbf{A}_{.j}\|_2\right)\left(\sum_{j=1}^bv_j\right)=\|\mathbf{A}\|_{2,\infty}\|v\|_1\tag{31}$$

Substituting (30) into (29) and then applying inequality (31) gives us

$$\|\mathcal{P}_{\hat{V}_{\mathcal{I}}^{(\Pi_n)}}\Delta_w^n\|_2^2 \leq \frac{2}{\hat{s}_{r_n}^2}\Big(\|\hat{\mathbb{E}}[\mathbf{Y}_{\mathcal{I}}^{(\Pi_n),r_n}]\hat{\mathbf{w}}^n - \mathbb{E}[\mathbf{Y}^{(\Pi_n)}]\|_2^2 + \|\hat{\mathbb{E}}[\mathbf{Y}_{\mathcal{I}}^{(\Pi_n),r_n}] - \mathbb{E}[\mathbf{Y}_{\mathcal{I}}^{(\Pi_n)}]\|_{2,\infty}^2\|\tilde{\mathbf{w}}^n\|_1^2\Big)$$
(32)

Next, we bound $\|\hat{\mathbb{E}}[\mathbf{Y}_{\mathcal{I}}^{(\Pi_n),r_n}]\hat{\mathbf{w}}^n - \mathbb{E}[\mathbf{Y}^{(\Pi_n)}]\|_2^2$. To this end, note that Assumption 3.1 implies that

$$\begin{aligned}
&\|\hat{\mathbb{E}}[\mathbf{Y}_{\mathcal{I}}^{(\Pi_n),r_n}]\hat{\mathbf{w}}^n - \mathbf{Y}^{(\Pi_n)}\|_2^2 \\
&= \|\hat{\mathbb{E}}[\mathbf{Y}_{\mathcal{I}}^{(\Pi_n),r_n}]\hat{\mathbf{w}}^n - \mathbb{E}[\mathbf{Y}^{(\Pi_n)}] - \boldsymbol{\epsilon}^{\Pi_n}\|_2^2 \\
&= \|\hat{\mathbb{E}}[\mathbf{Y}_{\mathcal{I}}^{(\Pi_n),r_n}]\hat{\mathbf{w}}^n - \mathbb{E}[\mathbf{Y}^{(\Pi_n)}]\|_2^2 + \|\boldsymbol{\epsilon}^{\Pi_n}\|_2^2 - 2\langle\hat{\mathbb{E}}[\mathbf{Y}_{\mathcal{I}}^{(\Pi_n),r_n}]\hat{\mathbf{w}}^n - \mathbb{E}[\mathbf{Y}^{(\Pi_n)}],\boldsymbol{\epsilon}^{\Pi_n}\rangle
\end{aligned}$$
(33)

Next, we proceed by calling upon Property 4.1 of [4] which states that $\hat{\mathbf{w}}^n$ as given by (4) is the unique solution to the following convex program:

$$\begin{aligned}
\min_{\mathbf{w}\in\mathbb{R}^{|\mathcal{I}|}}\quad & \|\mathbf{w}\|_2 \\
\text{s.t. } \mathbf{w}\in\operatorname*{argmin}_{\mathbf{w}\in\mathbb{R}^{|\mathcal{I}|}} & \|\mathbf{Y}^{(\Pi_n)} - \hat{\mathbb{E}}[\mathbf{Y}_{\mathcal{I}}^{(\Pi_n),r_n}]\mathbf{w}\|_2^2
\end{aligned}$$
(34)

Using this property, we have that

$$\begin{aligned}
&\|\hat{\mathbb{E}}[\mathbf{Y}_{\mathcal{I}}^{(\Pi_n),r_n}]\hat{\mathbf{w}}^n - \mathbf{Y}^{(\Pi_n)}\|_2^2 \\
&\leq \|\hat{\mathbb{E}}[\mathbf{Y}_{\mathcal{I}}^{(\Pi_n),r_n}]\tilde{\mathbf{w}}^n - \mathbf{Y}^{(\Pi_n)}\|_2^2 \\
&= \|\hat{\mathbb{E}}[\mathbf{Y}_{\mathcal{I}}^{(\Pi_n),r_n}]\tilde{\mathbf{w}}^n - \mathbb{E}[\mathbf{Y}^{(\Pi_n)}] - \boldsymbol{\epsilon}^{\Pi_n}\|_2^2 \\
&= \|\hat{\mathbb{E}}[\mathbf{Y}_{\mathcal{I}}^{(\Pi_n),r_n}]\tilde{\mathbf{w}}^n - \mathbb{E}[\mathbf{Y}_{\mathcal{I}}^{(\Pi_n)}]\tilde{\mathbf{w}}^n - \boldsymbol{\epsilon}^{\Pi_n}\|_2^2 \\
&= \|(\hat{\mathbb{E}}[\mathbf{Y}_{\mathcal{I}}^{(\Pi_n),r_n}] - \mathbb{E}[\mathbf{Y}_{\mathcal{I}}^{(\Pi_n)}])\tilde{\mathbf{w}}^n\|_2^2 + \|\boldsymbol{\epsilon}^{\Pi_n}\|_2^2 - 2\langle(\hat{\mathbb{E}}[\mathbf{Y}_{\mathcal{I}}^{(\Pi_n),r_n}] - \mathbb{E}[\mathbf{Y}_{\mathcal{I}}^{(\Pi_n)}])\tilde{\mathbf{w}}^n,\boldsymbol{\epsilon}^{\Pi_n}\rangle
\end{aligned}$$
(35)

Substituting (33) and (35) into (32) and using (31), we get

$$\begin{aligned}
&\|\hat{\mathbb{E}}[\mathbf{Y}_{\mathcal{I}}^{(\Pi_n),r_n}]\hat{\mathbf{w}}^n - \mathbb{E}[\mathbf{Y}^{(\Pi_n)}]\|_2^2 \\
&\leq \|(\hat{\mathbb{E}}[\mathbf{Y}_{\mathcal{I}}^{(\Pi_n),r_n}] - \mathbb{E}[\mathbf{Y}_{\mathcal{I}}^{(\Pi_n)}])\tilde{\mathbf{w}}^n\|_2^2 + 2\langle(\hat{\mathbb{E}}[\mathbf{Y}_{\mathcal{I}}^{(\Pi_n),r_n}])\Delta_w^n,\boldsymbol{\epsilon}^{\Pi_n}\rangle \\
&\leq \|(\hat{\mathbb{E}}[\mathbf{Y}_{\mathcal{I}}^{(\Pi_n),r_n}] - \mathbb{E}[\mathbf{Y}_{\mathcal{I}}^{(\Pi_n)}])\|_{2,\infty}^2\|\tilde{\mathbf{w}}^n\|_1^2 + 2\langle(\hat{\mathbb{E}}[\mathbf{Y}_{\mathcal{I}}^{(\Pi_n),r_n}])\Delta_w^n,\boldsymbol{\epsilon}^{\Pi_n}\rangle
\end{aligned}$$

Then substituting this equation into (32) gives us

$$\|\mathcal{P}_{\hat{V}_{\mathcal{I}}^{(\Pi_n)}}\Delta_w^n\|_2^2 \leq \frac{4}{\hat{s}_{r_n}^2}\Big(\|\hat{\mathbb{E}}[\mathbf{Y}_{\mathcal{I}}^{(\Pi_n),r_n}] - \mathbb{E}[\mathbf{Y}_{\mathcal{I}}^{(\Pi_n)}]\|_{2,\infty}^2\|\tilde{\mathbf{w}}^n\|_1^2 + \langle\hat{\mathbb{E}}[\mathbf{Y}_{\mathcal{I}}^{(\Pi_n),r_n}]\Delta_w^n,\boldsymbol{\epsilon}^{\Pi_n}\rangle\Big)$$
(36)

We state three lemmas that help us bound the equation above with their proofs given in Appendices D.3.1, D.3.2 and D.3.3 respectively.

**Lemma D.2.** *Let the set-up of Theorem 6.6 hold. Then,*

$$\hat{s}_{r_n} - s_{r_n} = O_p(1)$$

**Lemma D.3.** *Let the set-up of Theorem 6.6 hold. Then for a universal constant $C > 0$, we have*

$$\|\hat{\mathbb{E}}[\mathbf{Y}_{\mathcal{I}}^{(\Pi_n),r_n}] - \mathbb{E}[\mathbf{Y}_{\mathcal{I}}^{(\Pi_n)}]\|_{2,\infty} \leq C\sqrt{r_n|\Pi_n|}\Delta_E$$
(37)

**Lemma D.4.** *Let the set-up of Theorem 6.6 hold. Then,*

$$\langle\hat{\mathbb{E}}[\mathbf{Y}_{\mathcal{I}}^{(\Pi_n),r_n}]\Delta_w^n,\boldsymbol{\epsilon}^{\Pi_n}\rangle = O_p\Big(\sqrt{|\Pi_n|} + r_n + \|\hat{\mathbb{E}}[\mathbf{Y}_{\mathcal{I}}^{(\Pi_n),r_n}] - \mathbb{E}[\mathbf{Y}_{\mathcal{I}}^{(\Pi_n)}]\|_{2,\infty}\|\tilde{\mathbf{w}}^n\|_1\Big)$$
(38)

Using the results of three lemmas above, applying Assumption 6.4 in (36), and then simplifying gives us

$$\|\mathcal{P}_{\hat{V}_{\mathcal{I}}^{(\Pi_n)}}\Delta_w^n\|_2 = O_p\left(\frac{r_n\Delta_E\|\tilde{\mathbf{w}}^n\|_1}{\sqrt{|\mathcal{I}|}} + \frac{r_n\Big(1 + \sqrt{\Delta_E\|\tilde{\mathbf{w}}^n\|_1}\Big)}{\sqrt{|\mathcal{I}|}|\Pi_n|^{1/4}}\right)$$
(39)

This concludes the proof for term 2. Using the result of Lemma C.3, we have that $\|\tilde{\mathbf{w}}^n\|_1 \leq \sqrt{r_n}$. Substituting this into (39) gives us

$$\|\mathcal{P}_{\hat{V}_{\mathcal{I}}^{(\Pi_n)}} \Delta_w^n\|_2 = O_p\left( \frac{r_n^{3/2}\Delta_E}{\sqrt{|\mathcal{I}|}} + \frac{r_n\left(1 + \sqrt{\Delta_E}r_n^{1/4}\right)}{\sqrt{|\mathcal{I}|}|\Pi_n|^{1/4}} \right) \qquad (40)$$

*Collecting Terms.* Combining the results of (28) and (40), gives us

$$\|\mathcal{P}_{V_{\mathcal{I}}^{(\Pi_n)}} \Delta_w^n\|_2$$

$$= \tilde{O}_p\left( r_n^2\left[ \frac{\Delta_E}{\sqrt{|\mathcal{I}|}\min\{\sqrt{|\Pi_n|}, \sqrt{|\mathcal{I}|}\}} + \Delta_E^2 \right] + \frac{r_n^{3/2}\Delta_E}{\sqrt{|\mathcal{I}|}} + \frac{r_n\left(1 + \sqrt{\Delta_E}r_n^{1/4}\right)}{\sqrt{|\mathcal{I}|}|\Pi_n|^{1/4}} \right)$$

### D.3.1 Proof of Lemma D.2

We first state Weyl's inequality which will be useful for us to establish the results.

**Theorem D.5** (Weyl's Inequality)**.** *Given two matrices* $\mathbf{A}, \mathbf{B} \in \mathbb{R}^{m \times n}$*, let* $s_i$ *and* $\hat{s}_i$ *denote the i-th singular values of* $\mathbf{A}$ *and* $\mathbf{B}$ *respectively. Then, for all,* $i \leq \min\{n, m\}$*, we have* $|s_i - \hat{s}_i| \leq \|\mathbf{A} - \mathbf{B}\|_{op}$

Using Weyl's inequality gives us

$$|\hat{s}_{r_n} - s_{r_n}| \leq \|\mathbb{E}[\mathbf{Y}_{\mathcal{I}}^{(\Pi_n)}] - \hat{E}[\mathbf{Y}_{\mathcal{I}}^{(\Pi_n)}]\|_{op}$$
$$\leq \sqrt{|\mathcal{I}||\Pi_n|}\Delta_E$$

Using the inequality above and Assumption 6.4, we have

$$\hat{s}_{r_n} \geq s_{r_n} - \sqrt{|\mathcal{I}||\Pi_n|}\Delta_E$$
$$= s_{r_n}\left( 1 - \frac{\sqrt{|\mathcal{I}||\Pi_n|}\Delta_E}{s_{r_n}} \right)$$
$$\geq s_{r_n}(1 - \sqrt{r_n}\Delta_E)$$

Then, substituting (21) into the equation above gives us that

$$\frac{\hat{s}_{r_n}}{s_{r_n}} \geq \left( 1 - C\sqrt{\frac{r_n s^2 p}{M}} \right)$$

holds with high probability for some universal constant $C \geq 0$. Finally, using the assumption that $M = \omega(r_n s^2 p)$ yields the claimed result.

### D.3.2 Proof of Lemma D.3

For notational simplicity, let $\hat{\mathbf{U}}_{r_n}, \hat{\mathbf{S}}_{r_n}, \hat{\mathbf{V}}_{r_n}$ denote $\hat{\mathbf{U}}_{\mathcal{I}}^{(\Pi_n)}, \hat{\mathbf{S}}_{\mathcal{I}}^{(\Pi_n)}, \hat{\mathbf{V}}_{\mathcal{I}}^{(\Pi_n)}$ respectively. For a matrix $\mathbf{A}$, let $A_{.,j}$ denote its $j$-th column. Additionally, denote $\Delta_j = \hat{\mathbb{E}}[\mathbf{Y}_{\mathcal{I}}^{(\Pi_n),r_n}]_{.,j} - \mathbb{E}[\mathbf{Y}_{\mathcal{I}}^{(\Pi_n)}]_{.,j}$. Then,

$$\hat{\mathbb{E}}[\mathbf{Y}_{\mathcal{I}}^{(\Pi_n),r_n}]_{.,j} - \mathbb{E}[\mathbf{Y}_{\mathcal{I}}^{(\Pi_n)}]_{.,j}$$
$$= \left( \hat{\mathbb{E}}[\mathbf{Y}_{\mathcal{I}}^{(\Pi_n),r_n}]_{.,j} - \hat{\mathbf{U}}_{r_n}\hat{\mathbf{U}}_{r_n}^T \mathbb{E}[\mathbf{Y}_{\mathcal{I}}^{(\Pi_n)}]_{.,j} \right) + \left( \hat{\mathbf{U}}_{r_n}\hat{\mathbf{U}}_{r_n}^T \mathbb{E}[\mathbf{Y}_{\mathcal{I}}^{(\Pi_n)}]_{.,j} - \mathbb{E}[\mathbf{Y}_{\mathcal{I}}^{(\Pi_n)}]_{.,j} \right)$$

We have that $\left( \hat{\mathbb{E}}[\mathbf{Y}_{\mathcal{I}}^{(\Pi_n),r_n}]_{.,j} - \hat{\mathbf{U}}_{r_n}\hat{\mathbf{U}}_{r_n}^T \mathbb{E}[\mathbf{Y}_{\mathcal{I}}^{(\Pi_n)}]_{.,j} \right)$ belongs to the subspace spanned by the column vectors of $\hat{U}_{r_n}$, while $\left( \hat{\mathbf{U}}_{r_n}\hat{\mathbf{U}}_{r_n}^T \mathbb{E}[\mathbf{Y}_{\mathcal{I}}^{(\Pi_n)}]_{.,j} - \mathbb{E}[\mathbf{Y}_{\mathcal{I}}^{(\Pi_n)}]_{.,j} \right)$ belongs to its orthogonal complement. Therefore,

$$\|\hat{\mathbb{E}}[\mathbf{Y}_{\mathcal{I}}^{(\Pi_n),r_n}]_{.,j} - \mathbb{E}[\mathbf{Y}_{\mathcal{I}}^{(\Pi_n)}]_{.,j}\|_2^2$$
$$= \left\| \hat{\mathbb{E}}[\mathbf{Y}_{\mathcal{I}}^{(\Pi_n),r_n}]_{.,j} - \hat{\mathbf{U}}_{r_n}\hat{\mathbf{U}}_{r_n}^T \mathbb{E}[\mathbf{Y}_{\mathcal{I}}^{(\Pi_n)}]_{.,j} \right\|_2^2 + \left\| \hat{\mathbf{U}}_{r_n}\hat{\mathbf{U}}_{r_n}^T \mathbb{E}[\mathbf{Y}_{\mathcal{I}}^{(\Pi_n)}]_{.,j} - \mathbb{E}[\mathbf{Y}_{\mathcal{I}}^{(\Pi_n)}]_{.,j} \right\|_2^2$$

*Bounding* $\left\|\hat{\mathbb{E}}[\mathbf{Y}_{\mathcal{I}}^{(\Pi_n),r_n}]_{.,j} - \hat{\mathbf{U}}_{r_n}\hat{\mathbf{U}}_{r_n}^T\mathbb{E}[\mathbf{Y}_{\mathcal{I}}^{(\Pi_n)}]_{.,j}\right\|_2^2$. Observe that, we have

$$\hat{\mathbf{U}}_{r_n}\hat{\mathbf{U}}_{r_n}^T\hat{\mathbb{E}}[\mathbf{Y}_{\mathcal{I}}^{(\Pi_n)}]_{.,j} = \hat{\mathbf{U}}_{r_n}\hat{\mathbf{U}}_{r_n}^T\hat{\mathbb{E}}[\mathbf{Y}_{\mathcal{I}}^{(\Pi_n)}]\mathbf{e_j} = \hat{\mathbf{U}}_{r_n}\hat{\mathbf{U}}_{r_n}^T\hat{\mathbf{U}}_{\mathcal{I}}^{(\Pi_n)}\hat{\mathbf{S}}_{\mathcal{I}}^{(\Pi_n)}\hat{\mathbf{V}}_{\mathcal{I}}^{(\Pi_n)}\mathbf{e_j}$$
$$= \hat{\mathbf{U}}_{r_n}\hat{\mathbf{S}}_{r_n}\hat{\mathbf{V}}_{r_n}^T\mathbf{e}_j = \hat{\mathbb{E}}[\mathbf{Y}_{\mathcal{I}}^{(\Pi_n),r_n}]_{.,j}$$

Therefore, we have

$$\left\|\hat{\mathbb{E}}[\mathbf{Y}_{\mathcal{I}}^{(\Pi_n),r_n}]_{.,j} - \hat{\mathbf{U}}_{r_n}\hat{\mathbf{U}}_{r_n}^T\mathbb{E}[\mathbf{Y}_{\mathcal{I}}^{(\Pi_n)}]_{.,j}\right\|_2^2 = \left\|\hat{\mathbf{U}}_{r_n}\hat{\mathbf{U}}_{r_n}^T\hat{\mathbb{E}}[\mathbf{Y}_{\mathcal{I}}^{(\Pi_n)}]_{.,j} - \hat{\mathbf{U}}_{r_n}\hat{\mathbf{U}}_{r_n}^T\mathbb{E}[\mathbf{Y}_{\mathcal{I}}^{(\Pi_n)}]_{.,j}\right\|_2^2$$
$$\leq \left\|\hat{\mathbf{U}}_{r_n}\hat{\mathbf{U}}_{r_n}^T\right\|_2^2\left\|\hat{\mathbb{E}}[\mathbf{Y}_{\mathcal{I}}^{(\Pi_n)}]_{.,j} - \mathbb{E}[\mathbf{Y}_{\mathcal{I}}^{(\Pi_n)}]_{.,j}\right\|_2^2$$
$$\leq |\Pi_n|\Delta_E^2$$

*Bounding* $\left\|\hat{\mathbf{U}}_{r_n}\hat{\mathbf{U}}_{r_n}^T\mathbb{E}[\mathbf{Y}_{\mathcal{I}}^{(\Pi_n)}]_{.,j} - \mathbb{E}[\mathbf{Y}_{\mathcal{I}}^{(\Pi_n)}]_{.,j}\right\|_2^2$. Note that $\mathbb{E}[\mathbf{Y}_{\mathcal{I}}^{(\Pi_n)}]_{.,j} = \mathbf{U}_{\mathcal{I}}^{(\Pi_n)}\left(\mathbf{U}_{\mathcal{I}}^{(\Pi_n)}\right)^T\mathbb{E}[\mathbf{Y}_{\mathcal{I}}^{(\Pi_n)}]_{.,j}$. Using Assumption 6.1 and (26), we have

$$\left\|\hat{\mathbf{U}}_{r_n}\hat{\mathbf{U}}_{r_n}^T\mathbb{E}[\mathbf{Y}_{\mathcal{I}}^{(\Pi_n)}]_{.,j} - \mathbb{E}[\mathbf{Y}_{\mathcal{I}}^{(\Pi_n)}]_{.,j}\right\|_2^2 \leq \left\|\hat{\mathbf{U}}_{r_n}\hat{\mathbf{U}}_{r_n}^T - \mathbf{U}_{\mathcal{I}}^{(\Pi_n)}\left(\mathbf{U}_{\mathcal{I}}^{(\Pi_n)}\right)^T\right\|_2^2\|\mathbb{E}[\mathbf{Y}_{\mathcal{I}}^{(\Pi_n)}]_{.,j}\|_2^2$$
$$\leq \left\|\hat{\mathbf{U}}_{r_n}\hat{\mathbf{U}}_{r_n}^T - \mathbf{U}_{\mathcal{I}}^{(\Pi_n)}\left(\mathbf{U}_{\mathcal{I}}^{(\Pi_n)}\right)^T\right\|_2^2|\Pi_n|$$
$$\leq Cr_n|\Pi_n|\Delta_E^2$$

for a universal constant $C > 0$. Combining the two bounds gives us,

$$\max_j \Delta_j \leq C\sqrt{r_n|\Pi_n|}\Delta_E$$

for a universal constant $C > 0$.

### D.3.3  Proof of Lemma D.4

The proof of Lemma D.4 follows from adapting the notation of Lemma H.5 in [5] to that used in this paper. Specifically, we have the notational changes established in Table 2, as well as the following changes: $r_{pre} = r_n, \mathbf{Y}_{pre,\mathcal{I}^{(d)}}^{r_{pre}} = \hat{\mathbb{E}}[\mathbf{Y}_{\mathcal{I}}^{(\Pi_n),r_n}], \mathbb{E}[\mathbf{Y}_{pre,\mathcal{I}^{(d)}}], \epsilon_{pre,n} = \boldsymbol{\epsilon}^{\Pi_n}, T_0 = \Pi_n$ where the l.h.s of each equality is the notation used in [5], and the r.h.s is the notation used in this work. Using the notation established, we can use the result of Lemma H.5 in [5] to give us

$$\langle\hat{\mathbb{E}}[\mathbf{Y}_{\mathcal{I}}^{(\Pi_n),r_n}]\Delta_w^n, \boldsymbol{\epsilon}^{\Pi_n}\rangle = O_p\left(\sqrt{|\Pi_n|} + r_n + \|\hat{\mathbb{E}}[\mathbf{Y}_{\mathcal{I}}^{(\Pi_n),r_n}] - \mathbb{E}[\mathbf{Y}_{\mathcal{I}}^{(\Pi_n)}]\|_{2,\infty}\|\tilde{\mathbf{w}}^n\|_1\right)$$

which completes the proof.

### D.4  Proof of Proposition C.5

We prove Proposition C.5, which also serves as the finite-sample analysis of the vertical regression procedure (step 2 of Synthetic Combinations) done via principal component regression. The proof follows from adapting the notation of Proposition E.3 in [2] to that used in this paper. Specifically, we present table 2 that matches their notation to ours.

Using the table above, we have that the following holds with probability $1 - O\left((|\mathcal{I}||\Pi_n|)^{-10}\right)$

$$\|\hat{\mathbf{w}}^n - \tilde{\mathbf{w}}^n\|_2 \leq C\sigma^3\log^3((|\mathcal{I}||\Pi_n|))\left[\frac{\|\tilde{\mathbf{w}}^n\|_2}{\min\{\sqrt{|\Pi_n|}, \sqrt{|\mathcal{I}|}\}} + \Delta_E\right]$$

This completes the proof.

| Notation of [2] | Our Notation |
|:---:|:---:|
| $\mathbf{Y}$ | $\mathbf{Y}_{\Pi_n}$ |
| $\mathbf{X}$ | $\mathbb{E}[\mathbf{Y}_{\mathcal{I}}^{(\Pi_n)}]$ |
| $\mathbf{Z}$ | $\hat{\mathbb{E}}[\mathbf{Y}_{\mathcal{I}}^{(\Pi_n)}]$ |
| $n$ | $|\Pi_n|$ |
| $p$ | $|\mathcal{I}|$ |
| $\boldsymbol{\beta}^*$ | $\tilde{\mathbf{w}}$ |
| $\hat{\boldsymbol{\beta}}$ | $\hat{\mathbf{w}}$ |
| $\Delta_E$ | $\Delta_E$ |
| $r$ | $r$ |
| $\phi^{lr}$ | $0$ |
| $A$ | $1$ |
| $K$ | $0$ |
| $K_a, \kappa, \bar{\sigma}$ | $C\sigma$ |
| $\rho_{min}$ | $1$ |

Table 2: A summary of the main notational differences between our setting and that of [2].

# E  Sparsity of Matrix of Fourier Coefficients

In this section, we prove that the matrix of Fourier coefficients $\mathcal{A}$ has at most $rs$ non-zero columns. This result helps establish that the lower bound on the sample complexity for estimating all $N \times 2^p$ parameters is $O(Nr + r^2 s)$ as discussed in 6.3.

**Lemma E.1.** *The number of nonzero columns of $\mathcal{A}$, which we have denoted $p'$, satisfies $p' \leq rs$.*

*Proof.* Let $\mathcal{A}_t$ denote the sub-matrix of $\mathcal{A}$ formed by taking the first $t$ rows. Further, let $p'_t$ denote the number of nonzero columns of $\mathcal{A}_t$. We proceed by induction, that is we show for all $t$,

$$p'_t - s \cdot \mathrm{rank}(\mathcal{A}_t) \leq 0$$

In the case of $t = 1$, either the first row is the zero vector, in which case $p'_1 = \mathrm{rank}(\mathcal{A}_1) = 0$, or else $\mathrm{rank}(\mathcal{A}_1) = 1$ and $p'_1 \leq s$ since the first row is at most $s$-sparse. Then, for general $t$, note that

$$\mathrm{rank}(\mathcal{A}_{t-1}) \leq \mathrm{rank}(\mathcal{A}_t) \leq \mathrm{rank}(\mathcal{A}_{t-1}) + 1.$$

If $\mathrm{rank}(\mathcal{A}_t) = \mathrm{rank}(\mathcal{A}_{t-1})$ then we must also have $p'_{t-1} = p'_t$ and the inductive hypothesis holds. Otherwise, $\mathrm{rank}(\mathcal{A}_t) = \mathrm{rank}(\mathcal{A}_{t-1}) + 1$. Note that the $t^{\mathrm{th}}$ row of $\mathcal{A}$ has only $s$ nonzero entries, so $p'_t \leq p'_{t-1} + s$. In this case, we have that

$$\begin{aligned} p'_t - s \cdot \mathrm{rank}(\mathcal{A}_t) &= p'_t - s \cdot (\mathrm{rank}(\mathcal{A}_{t-1}) + 1) \\ &\leq (p'_{t-1} + s) - s \cdot (\mathrm{rank}(\mathcal{A}_{t-1}) + 1) \leq 0 \end{aligned}$$

where the last inequality holds due to the inductive hypothesis. Since $\mathrm{rank}(\mathcal{A}) = r$, we have that $p' \leq rs$. This completes the proof.

$\square$

