# OpenReview forum: "Synthetic Combinations: A Causal Inference Framework for Combinatorial Interventions"
_NeurIPS.cc/2023/Conference — NeurIPS 2023 poster_

### Official Review · Reviewer_z6nm · 2023-06-22

**Soundness:** 4 excellent
**Presentation:** 4 excellent
**Contribution:** 4 excellent
**Rating:** 7
**Confidence:** 4

**Summary:**

The manuscript introduces a synthetic combinations estimator, which computes causal effects for unseen treatment combinations under some reasonable assumptions on the data generating process. Building on theory from potential outcomes and synthetic interventions, the authors show how to exploit information sharing across units and treatments to identify causal effects in this challenging setting. The proposed two-step algorithm is more efficient and flexible than existing alternatives.

**Strengths:**

The topic is timely and interesting. The manuscript is exceptionally clear and well-written, which is greatly appreciated when dealing with dense formalisms. The theoretical results are strong and convincing, rooted in established results while simultaneously going beyond the current state of the art. The analysis is sound and well-motivated.

**Weaknesses:**

My main critique of this manuscript is one I am sure the authors will have anticipated – there are no empirical results in the main text! I am aware that 9 pages is tight but the authors could have been more judicious in their selection of what to send to the appendix. I also found the material on CART to be super interesting; a shame to banish this material to the wilderness of Appendices J and K. Fortunately, the final manuscript affords one extra page. I strongly encourage the authors to move Fig. 2 to the main text and expand on this empirical evaluation. Would also be great to shoehorn in some of the CART results, but that may prove difficult. If it is impossible to squeeze everything within the limit, then the authors may want to consider repurposing this manuscript for a journal submission. There is more than enough material here for a very solid journal contribution.

**Questions:**

See above.

---

> ### Author Rebuttal · Authors · 2023-08-08
>
> We thank the reviewer for their positive feedback, and address their concerns as follows.
>
> >My main critique of this manuscript is one I am sure the authors will have anticipated – there are no empirical results in the main text! I strongly encourage the authors to move Fig. 2 to the main text and expand on this empirical evaluation
>
> We thank the reviewer for highlighting this point about our presentation and empirical evaluation. We perform additional experiments in the global response on a real-world dataset on recommendation systems for combinations of movies that highlight the benefit of our approach as compared to other methods (e.g., Lasso and matrix completion techniques). Further, we show that the key assumptions required for Synthetic Combinations to work are satisfied in our real-data experiments. We will revise the paper to include these real-world experiments, as well as those in the appendix.
>
> > I also found the material on CART to be super interesting; a shame to banish this material to the wilderness of Appendices J and K. Fortunately, the final manuscript affords one extra page. Would also be great to shoehorn in some of the CART results, but that may prove difficult.
>
> We are glad the reviewer found the material on CART to be interesting! We highlight some results regarding CART in Corollary 6.8 which shows that CART can exploit additional regularity conditions placed on the potential outcomes to allow the sparsity $s$ to scale more quickly (i.e., by a factor of the number of interventions $p$) while achieving consistency. As a result, CART is able to achieve an improved sample complexity of $O(\text{poly}(r/\delta) \times (N + s^2))$ as compared to $O(\text{poly}(r/\delta) \times (N + s^2p))$ samples required when the horizontal regression is done via the Lasso. We will revise the paper to make the benefits of sample complexity of using CART clearer, and also attempt to include more formal results related to CART in the main text.

---

> > ### Author Response · Authors · 2023-08-14
> >
> > We thank the reviewer again for their thoughtful comments. We hope that they have had a chance to review our response to their specific concerns, and our real-world experiments in the global response where we demonstrate the efficacy of Synthetic Combinations over baselines methods, and that our key modeling assumptions (i.e., low-rank and sparsity) hold. Please let us know if there is anything else we can do to address your concerns, and we hope you improve your score.

---

> > > ### Comment · Reviewer_z6nm · 2023-08-14
> > > **Re: rebuttal**
> > >
> > > Thanks for your comments. I still believe my original score of 7 is fair and will keep it as such. Great work on this paper!

---

### Official Review · Reviewer_rsAG · 2023-07-03

**Soundness:** 3 good
**Presentation:** 2 fair
**Contribution:** 4 excellent
**Rating:** 7
**Confidence:** 3

**Summary:**

The paper studies the problem of estimating potential outcomes in the presence of a combinatorial number of intervention choices. Under some assumptions, they propose a two-phased algorithm "Synthetic Combinations": first exploit structure across combinations of interventions (via "horizontal regression") and then exploit structure across units (via "vertical regression"). Experiments are given in the appendix.

**Strengths:**

The proposed algorithm is clean and intuitive. It also seems to scale nicely with the number of intervention combinations in experiments.

While the theoretical guarantees rely heavily on a bunch of assumptions, Section 7 proposes an experimental design framework which ensures that an important set of assumptions (existence of donor units) will be met with high probability. In fact, I strongly propose that the authors rephrase their paper to highlight this; otherwise it is hard to believe that their work will useable as it is highly unlikely that all the required assumptions are met in practice without having control in assigning interventions to the units.

**Weaknesses:**

I did not check all the proofs in detail, but I do not see any glaring weaknesses.

There are a lot of assumptions and it is highly unlikely that all the required assumptions are met in practice (Section 7 helps to mitigate some of these concerns).

I am skeptical about the low-rank assumption on the matrix of Fourier coefficients $A$. While it is true low-rank assumptions are common in prior matrix completion settings, they usually directly consider the matrix at hand and not transform it into the Fourier space first. For example, Lines 655-657 in the appendix writes "This missingness pattern where outcomes with larger absolute values are observed is common in applications such as recommendation engines, where we are only likely to observe ratings for combinations that users either strongly like or dislike". The corresponding missingness pattern in the problem studied here is on the $N$-by-$2^p$ matrix. It is unclear to me why it should be believable that the transformed space is low-rank. The authors ought to justify this, ideally with practical examples/settings, or risk diminishing the impact of their contributions.

**Questions:**

Line 83:
By "equivalent", do you mean that they proved equivalence between the two problems via reductions, or do you mean "equivalent" in a colloquial sense of the word?

Assumption 3.1:
As discussed in the weaknesses, it is unclear to me why this model is interesting or justified. Of course, this work can be appreciated under the restriction of this assumption, but it will greatly weaken the contributions. I am more than happy to increase my "contribution" score if the authors provide sufficient justification for the low-rank assumption.

Type on Line 183:
double "exists"

Motivating example on Line 190:
I don't understand why this motivates the existence of donor units when the paper has thus far repeatedly claim to allow unobserved confounding. If we allow interventions to be arbitrarily assigned to units, it is unclear why we should believe that donor units exist. The "correct" way to justify should be to say that there is an experimental design that ensures the existence of donor units, and then refer to Section 7.

Determining donor set on Line 246:
This feels very ad-hoc. As it is unlikely that donor units will exist if we allow arbitrary experiments, I feel that this paragraph could be removed once the authors reorder their paper to place more emphasis on the experimental design proposed in Section 7.

Subsection on Additional Assumptions:
I feel that "so-and-so also has such an assumption" is not sufficient discussion of assumptions. Firstly, "so-and-so" may have the assumptions under different contexts (e.g. see my complaint about low-rank assumption in the Weaknesses section) so it is unclear why such assumption is justified in the setting studied in this paper. Secondly, the discussion should explain "what goes wrong" if one particular assumption is violated, or why we should expect any particular assumption to hold in practice. As mentioned several times by now, one "partial fix" is to emphasize that experimental design of Section 7 guarantees some assumptions with high probability. That is, "Synthetic Control" should be used in conjunction with the experimental design proposed in Section 7.

**Limitations:**

Nil.

---

> ### Author Rebuttal · Authors · 2023-08-08
>
> We thank the reviewer for their constructive feedback, and address their concerns as follows.
> > I am skeptical about the low-rank assumption on the matrix of Fourier coefficients $\mathcal{A}$. While it is true low-rank assumptions are common in prior matrix completion settings, they usually directly consider the matrix at hand and not transform it into the Fourier space first.
>
> > `Assumption 3.1: As discussed in the weaknesses, it is unclear to me why this model is interesting or justified... __I am more than happy to increase my "contribution" score if the authors provide sufficient justification for the low-rank assumption__.
>
> The $2^p \times N$ matrix of potential outcomes $\mathbb{E}[\mathbf{Y}_N^{(\Pi)}]$ can be written as $\mathbb{E}[\mathbf{Y}_N^{(\Pi)}] = \mathbf{\chi}(\Pi) \mathcal{A}^T$, where $\mathbf{\chi}(\Pi) $ is the matrix of Fourier characteristics. Since $\mathbf{\chi}(\Pi) $ is an invertible matrix, $\text{rank}(\mathbb{E}[\mathbf{Y}_N^{(\Pi)}]) = \text{rank}(\mathcal{A})$. Hence, placing a low-rank assumption on the Fourier coefficients $\mathcal{A}$ is equivalent to placing a low rank-assumption on the matrix of outcomes $\mathbb{E}[\mathbf{Y}_N^{(\Pi)}]$.  As the reviewer points out themselves, placing low-rank structure on the outcomes $\mathbb{E}[\mathbf{Y}_N^{(\Pi)}]$ is common when studying matrix completion. We discuss this equivalence in line 140, but will make this point clearer in our revision.
> > Determining donor set on Line 246: This feels very ad-hoc. As it is unlikely that donor units will exist if we allow arbitrary experiments...
>
> This paragraph provides a data-driven method using cross-validation (CV) to identify donor units in observational settings. Our additional experiments in the global response on real-world data demonstrates that we outperform other baselines (e.g., Lasso and matrix completion methods), motivating the existence of a donor set in an observational setting. Moreover, we use the method laid out in this paragraph of CV to select the donor set in our real-world experiments, indicating that this approach can be used in practice. However, the reviewer correctly points out that a donor set is not guaranteed to exist under arbitrary/adversarial observation patterns. We will clarify this in our revision.
> > Motivating example on Line 190: I don't understand why this motivates the existence of donor units when the paper has thus far repeatedly claim to allow unobserved confounding...
>
> We provide examples that guarantee the existence of donor units in both an experimental (i.e., our motivating example) and an observational setting. In the motivating example, the reviewer correctly points out that our treatment assignment mechanism does not induce unobserved confounding. We will revise the language to reflect this. In the observational setting, we provide an example (deferred to Appendix C for space constraints) where these assumptions hold under a natural model of unobserved confounding. Specifically, our observational example consists of a treatment assignment where only outcomes with large absolute values are seen. This observation pattern is common in recommendation systems where we only observe ratings from users who strongly like or dislike a product. We will  provide a brief description of this example in our revision.
> >Subsection on Additional Assumptions: I feel that "so-and-so also has such an assumption" is not sufficient discussion of assumptions. Firstly, "so-and-so" may have the assumptions under different contexts (e.g. see my complaint about low-rank assumption in the Weaknesses section) so it is unclear why such assumption is justified in the setting studied in this paper. Secondly, the discussion should explain "what goes wrong" if one particular assumption is violated...
>
> Thank you for pointing out the role of assumptions in our work. We discuss assumptions from these previous works in order to build upon them, and provide context for our work. However, we agree with the reviewer that we can revise our language to reflect how we discuss our assumptions better. We will revise the text to say that in our analysis, if the assumptions that place unit-specific structure (e.g., sparsity or incoherence of Fourier characteristics) are not satisfied (which can be tested via CV), then the outcomes of the donor units cannot be accurately estimated via the Lasso.  In this case, alternative horizontal regression algorithms to estimate donor unit potential outcomes may be required instead. Similarly, if the assumptions that place structure across units (e.g., low-rank condition) do not hold (which again can be tested by examining the spectra of the matrix), then it is difficult to accurately transfer the outcomes of the donor set to the non-donor units. In the global response we verify that these assumptions do seem to hold in our real-world experiments. Further, we discuss why important applications such as factorial design experiments and recommendation systems naturally induce low-rank and sparse representations in Appendix B.  However, we will discuss the limitations of our approach and "what goes wrong" if these assumptions do not hold in our revision.
> > Line 83: By "equivalent", do you mean that they proved equivalence between the two problems via reductions, or do you mean "equivalent" in a colloquial sense of the word
>
> We mean colloquially in the sense that both problems can be cast as missing data problems. That is, the central task of both causal inference and matrix completion is to impute unobserved (i.e., missing) outcomes. Specifically, given a ``causal inference estimator'' for imputing missing potential outcomes, one can use such an estimator to directly impute missing entries in the appropriately defined matrix.
> Similarly, given a matrix completion estimator to impute missing entries in a matrix, one can then use it to impute missing potential outcomes.
> > Type on Line 183: double "exists"
>
> Thanks! We will fix this.

---

> > ### Comment · Reviewer_rsAG · 2023-08-10
> >
> > Thank you for your patience and effort to clear my doubts and misunderstandings. Also, I appreciate the effort that the authors took to perform additional experiments --- it must have been tough to do so in such a short period of time! I am very satisfied with the detailed response and have updated the scores accordingly :)
> >
> > Please kindly incorporate some of the discussions here into your revision. Thanks!

---

> > > ### Author Response · Authors · 2023-08-14
> > >
> > > Thank you for taking the time to comment in such detail on our paper, and for reading our response. Your feedback was very helpful in improving our paper, and we will include them in our revision. Thank you for increasing your score!

---

### Official Review · Reviewer_qb2F · 2023-07-07

**Soundness:** 3 good
**Presentation:** 4 excellent
**Contribution:** 3 good
**Rating:** 7
**Confidence:** 3

**Summary:**

The goal paper studies the problem of recovering $N\times 2^p$ unit-specific outcomes for N heterogeneous units and any combination of p possible interventions from small number of experiments and observations.
Prior to this work, the problem has been studied under assumptions of latent similarity or regularity in how combinations of interventions interact, as well as some other setups. In the setup when one assumes latent similarity, namely that the matrix of Fourier coefficients across units has rank <= r, the problem is reduced to matrix completion, and hence all causal outcomes can be recovered from $O(poly(r)\times (N+2^p))$ observations. In the case, when regularity in intervention interactions is assumed it is known that $O(Ns^2p)$ measurement are sufficient, where s is the sparcity parameter of coefficients in the Fourier expansion of the potential outcomes.
This paper studies the problem in the case when both latent similarity and intervention regularity is assumed. Under both assumptions the paper proposes an algorithm that recovers all $N\times 2^p$ causal outcomes from $O(N\times (s^2 + p))$ measurements.

**Strengths:**

The problem of estimating causal outcomes under a combination of interventions is a notoriously hard problem with various applications. One complication is that it is usually expensive\impossible to run many experiments to measure the effects caused by interventions. Hence, understanding how to set up experiment that will require the minimum amount of measurements is of great importance.
This paper proposes an algorithm, called Synthetic Combinatorics, that provably recovers all  $N\times 2^p$ unit-specific outcomes under $2^p$ interventions under a combination of two widely accepted assumptions from  $O(N\times (s^2 + p))$, which is a significant improvement over the prior work.

The paper also provides statistical estimates for the number of samples needed for every experiment to achieve desired accuracy of the recovery.

The paper is well-written and as far as I can judge is correct, though I did not read the proofs carefully.

**Weaknesses:**

It is not completely clear how realistic is the scenario when latent similarity and intervention regularity holds simultaneously. I can imagine that in some datasets one or the other may hold, while both assumptions at the same time may not hold. The paper will benefit significantly from experiments on real-world datasets, that can confirm that theoretical assumptions are realistic and we indeed see improvement in the number of measurements needed.

**Questions:**

Can you provide some intuition why you believe that the assumptions needed for Synthetic Cobminatorics to work are expect to hold for real-world datasets?

**Limitations:**

-

---

> ### Author Rebuttal · Authors · 2023-08-08
>
> We thank the reviewer for their positive feedback, and address their concerns as follows.
>
> >It is not completely clear how realistic is the scenario when latent similarity and intervention regularity holds simultaneously. I can imagine that in some datasets one or the other may hold, while both assumptions at the same time may not hold. The paper will benefit significantly from experiments on real-world datasets, that can confirm that theoretical assumptions are realistic and we indeed see improvement in the number of measurements needed.
>
> > Can you provide some intuition why you believe that the assumptions needed for Synthetic Cobminatorics to work are expect to hold for real-world datasets?
>
> We thank the reviewer for pointing out the role of assumptions in our work. We perform additional experiments in the global response on a real-world dataset on recommendation systems for combinations of movies that highlight the benefit of our approach as compared to other methods (e.g., Lasso and matrix completion techniques). We also show that the key modeling assumptions (e.g., low-rank structure and sparsity) hold in this real-world dataset. Further, the improved performance of our method as compared to other approaches, and in particular, the Lasso, imply the existence of a valid donor set. We hope that these experiments motivate the empirical utility of our methods, and that our theoretical assumptions are grounded. We will revise the paper to include these results.
>
> We also note that latent similarity is equivalent to placing a low-rank assumption on the matrix of potential outcomes $\mathbb{E}[\mathbf{Y}_N^{(\Pi)}]$.  That is, the $2^p \times N$ matrix of potential outcomes $\mathbb{E}[\mathbf{Y}_N^{(\Pi)}]$ can be written as $\mathbb{E}[\mathbf{Y}_N^{(\Pi)}] = \mathbf{\chi}(\Pi) \mathcal{A}^T$, where $\mathbf{\chi}(\Pi)$ is the matrix of Fourier characteristics. Since $\mathbf{\chi}(\Pi)$ is an invertible matrix, $\text{rank}(\mathbb{E}[\mathbf{Y}_N^{(\Pi)}] ) = \text{rank}(\mathcal{A})$. Hence, placing a low-rank assumption on the Fourier coefficients $\mathcal{A}$ is equivalent to placing a low rank-assumption on the matrix of outcomes $\mathbb{E}[\mathbf{Y}_N^{(\Pi)}]$, which is a widely made assumption when studying matrix completion. With regards to intervention regularity and sparsity, we provide a discussion of why these assumptions hold in models used to study relevant applications such as factorial design experiments and recommendation systems in Appendix B.

---

> > ### Author Response · Authors · 2023-08-14
> >
> > We thank the reviewer again for their thoughtful comments. We hope that they have had a chance to review our response to their specific concerns, and our real-world experiments in the global response where we demonstrate the efficacy of Synthetic Combinations over baselines methods, and that our key modeling assumptions (i.e., low-rank and sparsity) hold. Please let us know if there is anything else we can do to address your concerns, and we hope you improve your score.

---

### Official Review · Reviewer_Wabz · 2023-07-27

**Soundness:** 4 excellent
**Presentation:** 4 excellent
**Contribution:** 4 excellent
**Rating:** 7
**Confidence:** 1

**Summary:**

__Disclaimer__: This is my first time reading & reviewing a paper from the field of Combinatorial Interventions. My expertise is in NLP.

This work studies latent structure across units and combinations of interventions, assuming similar outcomes across units and regular interaction. An estimation procedure, Synthetic Combinations, is proposed, establishing finite-sample consistency under precise conditions. This work also uses methods to reduce errors in variables and provides a possibility of model-agnostic analysis.


**Strengths:**

* All the proofs and other mathematical explanations are clear, but I'm not able to understand them properly because of no expertise in that field.

**Weaknesses:**

*

**Questions:**

NA

---

> ### Author Rebuttal · Authors · 2023-08-08
>
> We thank the reviewer for their positive feedback! We note that we perform additional experiments in the global response on a real-world dataset on recommendation systems for combinations of movies that highlight the benefit of our approach as compared to other methods (e.g., Lasso and matrix completion techniques). Further, we show that the key assumptions required for Synthetic Combinations to work are satisfied in our real-data experiments. We hope that these experiments motivate the empirical utility of Synthetic Combinations, and we will revise the paper to include these results.

---

> > ### Author Response · Authors · 2023-08-14
> >
> > We thank the reviewer again for their thoughtful comments. We hope that they have had a chance to review our response to their specific concerns, and our real-world experiments in the global response where we demonstrate the efficacy of Synthetic Combinations over baselines methods, and that our key modeling assumptions (i.e., low-rank and sparsity) hold. Please let us know if there is anything else we can do to address your concerns, and we hope you improve your score.

---

### Author Rebuttal · Authors · 2023-08-08

We thank the reviewers for their positive feedback! A primary concern amongst reviewers was a lack of empirical evaluation
of Synthetic Combinations. Here, we present a real-world data experiment on recommendation systems for sets of movie
ratings to address these concerns, and highlight the empirical effectiveness of our approach. Further, we empirically validate
that the key assumptions (i.e., low-rank condition and sparsity of donor unit Fourier coefficients) required for Synthetic
Combinations to work also hold. We address specific reviewer concerns in rebuttals to each of them separately.

__Data and Experimental Set-up.__ We use data collected in [2] which consists of user ratings of sets of movies. Specifically, users were asked to provide a rating of 1-5 on a set of 5 movies chosen at random. This resulted in a total of ratings from 854 users over 29, 516 sets containing 12, 549 movies. More details about the data collection process can be found in [2]. Due to computational constraints, we only perform experiments on N = 100 users and 4000 sets of ratings chosen at random. We use 80% of each user’s ratings as the training set, and the other 20% as the test set to evaluate performance.

__Comparison Methods__. As in the numerical simulations in the appendix, we compare Synthetic Combinations to matrix
completion algorithms: SoftImpute [ 1], and IterativeSVD [ 3]. These methods require that the rank of the underlying
matrix be provided as a hyper-parameter. This was chosen via 5-fold cross-validation (CV). We also compare Synthetic
Combinations to the Lasso, where we tune the regularization parameter λ via 5-fold CV. For Synthetic Combinations, we
tune all hyper-parameters via 5-fold CV. Additionally, we choose the donor set via the approached outlined in the manuscript
(see lines 246-257).

__Results.__ We measure the root mean squared error (RMSE) for all methods, and average their results over 3 repetitions. The
RMSE is displayed in the table below. We observe that Synthetic Combinations outperforms all other methods. Further, the
gap between Synthetic Combinations and the Lasso shows the benefit of first estimating the outcomes of the donor set, and
then transferring these estimated outcomes to non-donor units. We hope that this experiment enforces the empirical utility
of our approach, and showcases that our theoretical and modeling assumptions are grounded. We will revise the manuscript
to include these experiments, and expand on this empirical evaluation as well (e.g., investigating performance as a function
of the number of users and sets).

| Method | __Synthetic Combinations__ | SoftImpute | IterativeSVD | Lasso           |
|--------|--------------------------|---------------------|-----------------------|-----------------|
| RMSE   | __0.55__ $\pm$ 0.06 | 0.67 $\pm$ 0.03 | 0.68 $\pm$ 0.02 | 0.80 $\pm$ 0.12 |

__Key Assumptions of Synthetic Combinations hold__. We also verify that two of the key assumptions, the low-rank
condition on the matrix of outcomes and sparsity of donor unit Fourier coefficients, hold in this real-world dataset. For the
low-rank condition, we choose the set of movies that were rated by all users, and and plot its singular value spectrum (using a log-scale for the magnitude of the spectrum) in Figure 1 of the attached PDF. As seen in the plot, it is clear that the matrix of outcomes displays low-rank structure. For the sparsity condition, we investigate the Lasso model that was learnt for the donor units. The RMSE averaged across all the donor units on the test set was 0.51, indicating that the estimated Fourier coefficient is an accurate representation of the true underlying Fourier coefficient. Further, we note that the estimated Fourier coefficients are indeed sparse, and on average only 8.7\% of all possible coefficients are non-zero.


__References__

[1] R. Mazumder, T. Hastie, and R. Tibshirani. Spectral regularization algorithms for learning large incomplete matrices.
The Journal of Machine Learning Research, 11:2287–2322, 2010.

[2] M. Sharma, F. M. Harper, and G. Karypis. Learning from sets of items in recommender systems. ACM Trans. Interact.
Intell. Syst., 9(4), jul 2019. ISSN 2160-6455. doi: 10.1145/3326128. URL https://doi.org/10.1145/3326128.

[3] O. Troyanskaya, M. Cantor, G. Sherlock, P. Brown, T. Hastie, R. Tibshirani, D. Botstein, and R. B. Altman. Missing
value estimation methods for dna microarrays. Bioinformatics, 17(6):520–525, 2001.

---

### Decision · Program_Chairs · 2023-09-21

**Decision:**

Accept (poster)

**Comment:**

This paper proposes a provably intervention-efficient method for learning potential outcomes for any combination of p interventions. This is a challenging problem, and the reviewers have unanimously recommended acceptance. Please incorporate the changes requested by the authors in the camera ready revision.